# SPATIAL FORCING: IMPLICIT SPATIAL REPRESENTATION ALIGNMENT FOR VISION-LANGUAGE-ACTION MODEL

**Fuhao Li**[1,2,*] **Wenxuan Song**[1,*,†] **Han Zhao**[3,4] **Jingbo Wang**[1,5] **Pengxiang Ding**[3,4]
**Donglin Wang**[3] **Long Zeng**[2,‡] **Haoang Li**[1,‡]
[1]The Hong Kong University of Science and Technology (Guangzhou)  [2]Tsinghua University
[3]Westlake University  [4]Zhejiang University  [5]South China University of Technology

## ABSTRACT

Vision-language-action (VLA) models have recently shown strong potential in enabling robots to follow language instructions and execute precise actions. However, most VLAs are built upon vision-language models pretrained solely on 2D data, which lack accurate spatial awareness and hinder their ability to operate in the 3D physical world. Existing solutions attempt to incorporate explicit 3D sensor inputs such as depth maps or point clouds, but these approaches face challenges due to sensor noise, hardware heterogeneity, and incomplete depth coverage in existing datasets. Alternative methods that estimate 3D cues from 2D images also suffer from the limited performance of depth estimators. We propose Spatial Forcing (SF), a simple yet effective alignment strategy that implicitly forces VLA models to develop spatial comprehension capabilities without relying on explicit 3D inputs or depth estimators. SF aligns intermediate visual embeddings of VLAs with geometric representations produced by pretrained 3D foundation models. By enforcing alignment at intermediate layers, SF guides VLAs to encode richer spatial representations that enhance action precision. Extensive experiments in simulation and real-world environments demonstrate that SF achieves state-of-the-art results, surpassing both 2D- and 3D-based VLAs. SF further accelerates training by up to 3.8× and improves data efficiency across diverse robotic tasks. Project page: https://spatial-forcing.github.io/.

## 1 INTRODUCTION

The development of robotic manipulation hinges on integrating semantic reasoning with accurate physical control in the 3D real world. Recently, vision-language-action (VLA) models (Brohan et al., 2022; Zitkovich et al., 2023; Bjorck et al., 2025; Black et al., 2024), capable of instruction following and robotic action execution, have attracted significant attention. Most VLA models are built upon vision-language models (VLMs) (Liu et al., 2023b; Chen et al., 2023; Karamcheti et al., 2024) to inherit semantic understanding capabilities and employ action tokenization (Kim et al., 2024; Song et al., 2025b) and action experts (Wen et al., 2025; Li et al., 2024a) to output actions.

However, the VLM backbones of these 2D VLA models are pretrained solely on 2D visual modalities and lack precise spatial awareness (Wang et al., 2025c), making it infeasible to adapt them to the 3D physical world. To address this, recent 3D VLAs have explored incorporating 3D modalities into inputs to exploit the rich geometric priors of the physical environment for precise manipulation. Several approaches (Chen et al., 2025; Li et al., 2025a; Bhat et al., 2025a; Li et al., 2025b; Sun et al., 2025) use depth cameras or lidars to obtain depth maps or point clouds as input, which are then encoded and explicitly fed into the VLM backbone or diffusion head. However, some limitations make it difficult to develop a universal and scalable 3D strategy for this paradigm: (1) Reliable 3D

---

*Equal Contribution.

†Project Lead: songwenxuan0115@gmail.com

‡Corresponding Authors: haoangli@hkust-gz.edu.cn and zenglong@sz.tsinghua.edu.cn

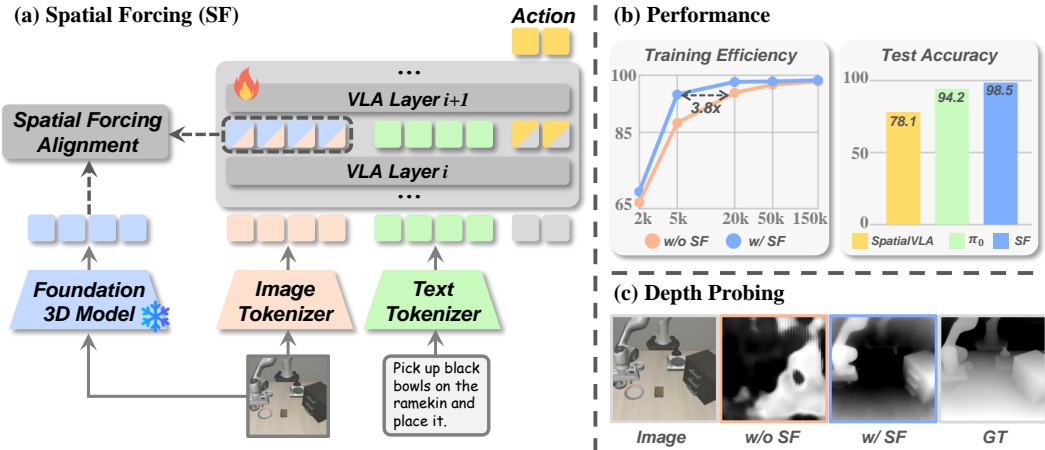

Figure 1: **Our proposed method, Spatial Forcing (SF), implicitly forces VLA models to acquire spatial-aware knowledge. (a)** SF aligns intermediate visual embeddings of VLAs with geometric representations from pretrained 3D foundation models. **(b)** Our simple yet effective strategy yields significant improvements in training efficiency and test accuracy. **(c)** Depth probing proves that our SF brings spatial information into the aligned representations, further enhancing 3D perception.

sensor data is hard to acquire, as depth maps and point clouds obtained from sensors are often low-quality and inaccurate. (2) Robotic sensors vary in types, positions, and calibration status, which introduces substantial heterogeneity into the data. (3) Portions of existing large-scale robot datasets (O'Neill et al., 2024b; Jang et al., 2022) do not include depth information, which limits the potential of scaling. Other approaches (Qu et al., 2025) attempt to estimate 3D information from 2D images using depth estimators. Yet, their effectiveness is inherently limited by the performance of the depth estimator, which finally yields sub-optimal performance. Therefore, an essential question for this field now is **how to implicitly develop VLAs' 3D perception and comprehension capabilities, thereby eliminating the dependence on explicit 3D sensor information or depth estimators?**

To analyze this, we conduct a lightweight depth probing experiment: we extract the frozen visual embeddings from a mainstream VLA (Kim et al., 2025) and only train a DPT head (Ranftl et al., 2021) to predict the depth maps. As shown in Fig. 1(c), this observation proves that the original visual embeddings fail to yield meaningful spatial structures, revealing a potential gap in the spatial reasoning capabilities of VLAs trained without external guidance. To bridge the gap, we propose **Spatial Forcing (SF)**, a simple yet effective alignment strategy that implicitly forces VLA models to acquire spatial-aware knowledge. Specifically, for auto-regressive VLAs with causal attention, action tokens are generated conditioning on the preceding visual and linguistic tokens. Intuitively, higher-quality visual tokens that contain richer spatial information can facilitate generating more precise manipulation actions. Recent advances (Huang et al., 2025; Wang et al., 2024; Yu et al., 2024) have demonstrated the effectiveness of representation supervision. Inspired by this, as shown in Fig. 1(a), we align intermediate visual embeddings of VLAs with external spatial representations extracted from pretrained 3D foundation models. Technically, to ensure multi-view consistency, we employ VGGT (Wang et al., 2025b) as the 3D foundation model to simultaneously process robotic images and generate normalized spatial representations, which serve as supervision signals. Overall, by aligning representations, we implicitly force VLAs to develop 3D comprehension capabilities.

Experiments conducted across multiple simulation environments and the real world validate the effectiveness of SF. Results (Fig. 1(b)) on LIBERO and RoboTwin demonstrate that our SF achieves state-of-the-art (SOTA) performance, surpassing previous strong baselines, including 2D and 3D VLAs. Additional experiments on training iterations and dataset sizes indicate that our method realizes $3.8\times$ training while also exhibiting data efficiency with significantly less data. Finally, real-world experiments demonstrate the spatial comprehension and data utilization capabilities.

In summary, the contributions of this paper are threefold:

- We provide an observation based on depth probing, supported by an interpretable analysis, to highlight the insufficiency of spatial information in the visual embeddings of VLAs.

- We introduce Spatial Forcing, a simple yet effective alignment strategy that implicitly enforces the integration of visual embeddings in VLAs with external spatial representations.

- Experiments prove that our method demonstrates enhanced performance, accelerated training speeds, and improved data efficiency across diverse robotic tasks.

## 2 METHOD

### 2.1 PRELIMINARIES

**Vision-language-action Models.** VLA models are built upon pretrained VLMs and generate actions through specialized designs for action experts. A $\theta$-parameterized VLM model employs multiple causal attention layers in an auto-regressive manner to generate the next tokens, formed as $\boldsymbol{x}_t \sim p_\theta(\boldsymbol{x}_t \mid \boldsymbol{x}_{<t})$, where $\{\boldsymbol{x}_t\}_{t=1}^T$ denotes a sequence of tokens.

When adapted to VLA models, three modalities of information will be processed: vision, text, and action. The vision modality consists of multi-view images captured by robots, which are transformed into $N$ visual tokens $\{\boldsymbol{x}_t^{\mathcal{V}}\}_{t=1}^N$ through pretrained visual encoders such as DINOv2 (Oquab et al., 2023) or SigLIP (Zhai et al., 2023). The text modality consists of task instructions, which are converted into $M$ linguistic tokens $\{\boldsymbol{x}_t^{\mathcal{L}}\}_{t=1}^M$ by a text tokenizer. Then, the VLA models generate $K$ action tokens $\{\boldsymbol{x}_t^{\mathcal{A}}\}_{t=1}^K$ conditioned on the preceding visual and linguistic tokens:

$$\boldsymbol{x}_t^{\mathcal{A}} \sim p_\theta\big(\boldsymbol{x}_t^{\mathcal{A}} \mid \{\boldsymbol{x}_i^{\mathcal{V}}\}_{i=1}^N, \{\boldsymbol{x}_j^{\mathcal{L}}\}_{j=1}^M, \boldsymbol{x}_{<t}^{\mathcal{A}}\big), \tag{1}$$

$$\mathcal{L}_{\text{action}} = \mathcal{L}[\mathcal{G}(\{\boldsymbol{x}_t^{\mathcal{A}}\}_{t=1}^K), A_{gt}], \tag{2}$$

where $\mathcal{L}[\cdot, \cdot]$ denotes the training loss (*e.g.* L1, L2, or cross-entropy loss), $\mathcal{G}$ denotes the trainable action expert (*e.g.* two-layer MLP or flow-matching head (Lipman et al., 2022)). Eq. (1) illustrates that the visual tokens as intermediate scene representations play a crucial role in generating action tokens and could be supervised properly.

**Visual Geometry Grounded Transformer (VGGT).** VGGT (Wang et al., 2025b) is a feedforward model that directly outputs various 3D attributes of a scene, including camera parameters, point maps, depth maps, and 3D point tracks, based on a series of 2D images. It is composed of a transformer backbone and multiple prediction heads. To make the Transformer focus within each frame and globally in an alternate way, the model employs an Alternating-Attention mechanism that interleaves frame-wise self-attention and global self-attention. For each frame, local and global features are integrated into a unified latent representation, which is subsequently processed by a set of task-specific heads to produce corresponding 3D attributes. In our work, we argue that the latent representation extracted from the VGGT transformer backbone inherently encodes rich spatial information and is sufficient to serve as the 3D supervision signal.

### 2.2 MOTIVATION

**Challenge.** It is significantly challenging to bridge the gap between VLMs pretrained solely on 2D images and the underlying dynamic 3D structure of the physical world. Several approaches (Chen et al., 2025; Li et al., 2025a; Bhat et al., 2025a; Li et al., 2025b; Sun et al., 2025) (Fig. 2(a)) incorporate depth cameras to obtain depth maps as input, but the effectiveness is limited by low quality and limited quantity of 3D data. Other approaches (Qu et al., 2025; Lin et al., 2025) (Fig. 2(b)) attempt to estimate 3D information from 2D images, but their capabilities are limited by the performance of the depth estimators, which makes the policy sub-optimal. These challenges motivate our exploration of a universal and scalable training paradigm for 3D VLAs.

**Observation.** Since VLA models must capture 3D information and reason about the relative positions between robots and objects to generate precise spatial movements, we hypothesize that the 3D information is implicitly embedded within the visual embeddings of VLAs. Such embeddings would allow action tokens to acquire 3D cues through the auto-regressive mechanism during inference.

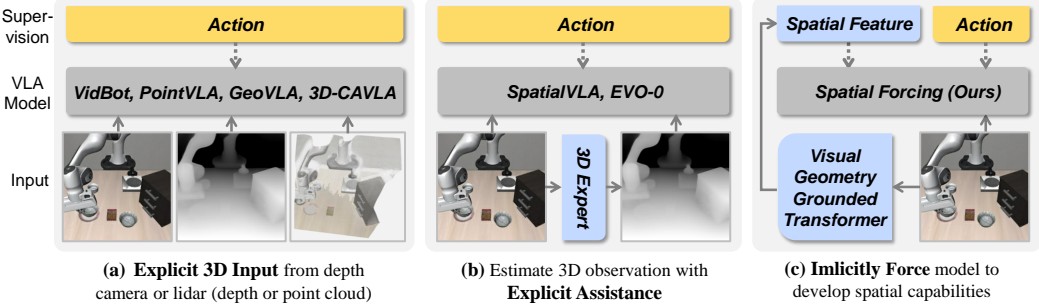

(a) **Explicit 3D Input** from depth camera or lidar (depth or point cloud)

(b) Estimate 3D observation with **Explicit Assistance**

(c) **Imlicitly Force** model to develop spatial capabilities

Figure 2: **Comparison among different paradigms for 3D VLAs.**

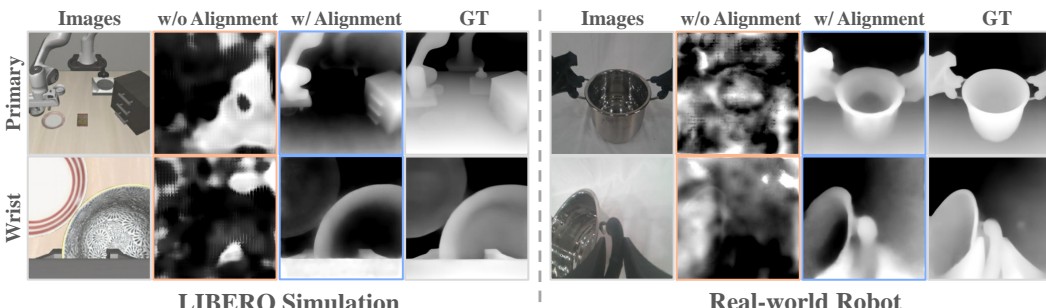

LIBERO Simulation

Real-world Robot

Figure 3: **Depth probing** of the visual embeddings of VLAs. Embeddings learned solely from 2D images without alignment do not produce meaningful spatial structures. The aligned embeddings inherently contain rich spatial information, leading to better performance in depth probing.

To evaluate this hypothesis, we follow linear probing (He et al., 2020) and Wu et al. (2025) to conduct a lightweight **depth probing** experiment. Specifically, we freeze all the parameters of a VLA model and only train a DPT head (Ranftl et al., 2021) to transform the visual embeddings of VLA to depth maps. This enables us to quantify the richness of spatial information embedded in the VLA representation space. As shown in Fig. 3, the probing results show that visual embeddings learned solely from 2D images do not produce meaningful spatial structures, suggesting a limited capacity of the model to encode 3D structure information without explicit spatial input or guidance.

## 2.3 SPATIAL FORCING

Our main philosophy is to employ external supervision signals to supervise the visual tokens $x^\mathcal{V}$. To provide the signals with rich spatial information, we first input a set of multi-view images $\mathcal{I}$ into the pretrained 3D foundation model VGGT $f^{3D}$, which outputs the pixel-level spatial representations $f^{3D}(I)$ for each image $I$. Additionally, these spatial representations are added with extra positional embedding $E$ to ensure that the supervised tokens preserve the critical position order within the auto-regressive process in the VLA. To align the VLA's per-pixel visual tokens with this 3D foundation model, we first process each $x_i$ with batch normalization $\Gamma$ followed by a two-layer MLP to ensure compatibility in feature dimension. Then we employ a cosine similarity score to maximize the alignment between the visual tokens of VLA and the spatial representation signals:

$$\mathcal{L}_{\text{align}} = -\frac{1}{N}\sum_{i=1}^{N}\mathcal{S}[\text{MLP}\cdot\Gamma(x_i^\mathcal{V}), f_i^{3D}(I) + E], \qquad (3)$$

where $S[\cdot,\cdot]$ denotes cosine similarity, $f_i^{3D}(I)$ is the spatial representations corresponding to the pixel location of visual token $x_i^\mathcal{V}$.

Additionally, as there are multiple causal attention layers in the VLA model, there are multiple $x^\mathcal{V}$ that can be gained after different layers. We found that supervising relatively deep but not the deepest layers is most effective in enhancing action performance. The reason is probably that the

Table 1: **Comparisons with state-of-the-art methods** on LIBERO benchmark. Please note that methods in gray font incorporate extra depth or point cloud information from other sensors. **Bold** denotes the best performances among the methods without extra sensor inputs.

| Method | Spatial SR (%) | Object SR (%) | Goal SR (%) | Long SR (%) | Average SR (%) |
|---|---|---|---|---|---|
| 2D VLA | | | | | |
| Diffusion Policy (Chi et al., 2023)*[RSS'23]* | 78.3 | 92.5 | 68.3 | 50.5 | 72.4 |
| TraceVLA (Zheng et al., 2025)*[ICLR'25]* | 84.6 | 85.2 | 75.1 | 54.1 | 74.8 |
| Octo (Ghosh et al., 2024)*[RSS'24]* | 78.9 | 85.7 | 84.6 | 51.1 | 75.1 |
| Openvla (Kim et al., 2024)*[CoRL'24]* | 84.7 | 88.4 | 79.2 | 53.7 | 76.5 |
| Dita (Hou et al., 2025)*[ICCV'25]* | 84.2 | 96.3 | 85.4 | 63.8 | 82.4 |
| CoT-VLA (Zhao et al., 2025)*[CVPR'25]* | 87.5 | 91.6 | 87.6 | 69.0 | 83.9 |
| $\pi_0$-FAST (Pertsch et al., 2025)*[RSS'25]* | 96.4 | 96.8 | 88.6 | 60.2 | 85.5 |
| $\pi_0$ (Black et al., 2024)*[RSS'25]* | 96.8 | 98.8 | 95.8 | 85.2 | 94.2 |
| UniVLA (Bu et al., 2025)*[RSS'25]* | 96.5 | 96.8 | 95.6 | 92.0 | 95.2 |
| Openvla-OFT (Kim et al., 2025)*[RSS'25]* | 97.6 | 98.4 | 97.9 | 94.5 | 97.1 |
| Explicit 3D VLA | | | | | |
| SpatialVLA (Qu et al., 2025)*[RSS'25]* | 88.2 | 89.9 | 78.6 | 55.5 | 78.1 |
| GeoVLA (Sun et al., 2025)*[arXiv'25]* | 98.4 | 99.0 | 96.6 | 96.6 | 97.7 |
| 3D-CAVLA (Bhat et al., 2025b)*[arXiv'25]* | 98.2 | 99.8 | 98.2 | 96.1 | 98.1 |
| Implicit 3D VLA | | | | | |
| Spatial Forcing (Ours) | **99.4** | **99.6** | **98.8** | **96.0** | **98.5** |

deeper layers lose more vision-specific features, making them less amenable to the supervision of target spatial representations, which is analogous to the conclusion in Huang et al. (2024).

The final training objective combines both the standard training loss for action generation and the 3D foundation model alignment loss with a weighting factor $\alpha$:

$$\mathcal{L}_{\text{SF}} = \mathcal{L}_{\text{action}} + \alpha \mathcal{L}_{\text{align}}. \tag{4}$$

Overall, through SF, the VLA model acquires spatial reasoning capabilities implicitly and efficiently.

**Model Inference.** During inference, the VLA model trained in the SF manner operates identically to a standard VLA without SF, introducing no additional structures or computational overhead, thereby highlighting SF's high applicability.

## 3 SIMULATION EXPERIMENTS

### 3.1 EXPERIMENTAL SETUP

**Simulation Environment.** We evaluate our method on two widely-used simulation benchmarks, LIBERO (Liu et al., 2023a) and RoboTwin (Mu et al., 2025). **LIBERO** consists of four main task suites: LIBERO-Spatial, LIBERO-Object, LIBERO-Goal, and LIBERO-Long. Each task suite contains 500 expert demonstrations across 10 tasks to investigate policy generalization to different spatial layouts, objects, goals, and long-horizon tasks. **RoboTwin** is a real-to-sim bimanual benchmark. It contains an easy setting with in-domain layout and a hard setting with domain randomization, including scene clutter, diverse background textures, lighting variation, and varied tabletop heights. We evaluate our methods on diverse tasks. We report the success rates (**SR**) as evaluation metrics for these two benchmarks.

**Base Models and Implementation Details.** We follow Kim et al. (2025) and Black et al. (2024) to employ OpenVLA-OFT on LIBERO, and $\pi_0$ on RoboTwin, as base models. **OpenVLA-OFT** uses the Prismatic VLM (Karamcheti et al., 2024) pretrained on the Open-X-Embodiedment dataset

(O'Neill et al., 2024b) as the VLM backbone and the fused vision backbone (both SigLIP (Zhai et al., 2023) and DINOv2 (Oquab et al., 2023)) as the vision backbone. We train our SF based on OpenVLA-OFT on 8 NVIDIA H100 for 150k iterations to compare with other methods. $\pi_0$ use the PaliGemma (Beyer et al., 2024) as VLM backbone. We train our SF based on $\pi_0$ with LoRA (Hu et al., 2022) on 1 NVIDIA H100 for 30k iterations. To ensure fairness, all training and evaluations follow the official settings.

## 3.2 COMPARISONS WITH STATE-OF-THE-ART METHODS

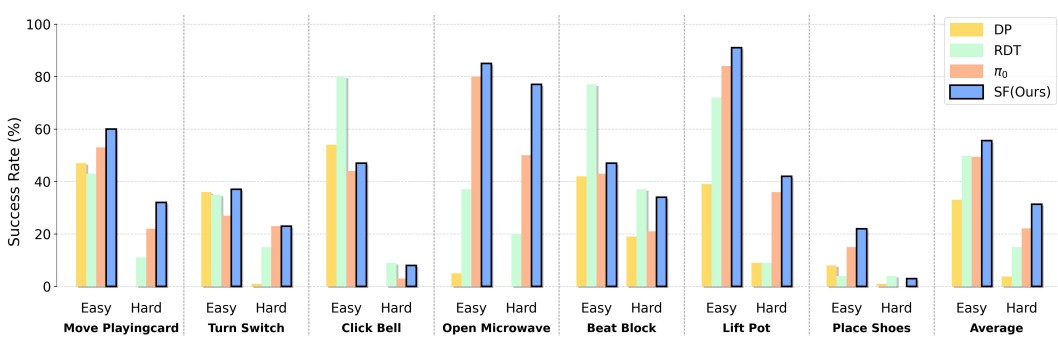

Figure 4: **Comparisons with state-of-the-art methods** on RoboTwin 2.0 benchmark.

**LIBERO.** Each task is evaluated for 500 trials under random seeds. Tab. 1 shows that SF gets the best performance across all four tasks. Specifically, strictly following the same setup of OpenVLA-OFT to use both primary and wrist camera images, our method outperforms them quite a lot. Besides, for long-horizon tasks on LIBERO-Long, our method also demonstrates strong capabilities to complete long-horizon tasks. Please note that, without requiring additional 3D inputs, our SF achieves comparable performance with those methods that benefit from extra 3D sensor inputs (*e.g.*, GeoVLA and 3D-CAVLA).

**RoboTwin.** Each easy task is evaluated for 100 trials, and each hard task is evaluated for 300 trials under random seeds. Fig. 4 shows that our SF achieves the highest average success rate and yields substantial improvements over the base model $\pi_0$ across all tasks, which demonstrates its effectiveness in enhancing the spatial awareness. Moreover, the obvious improvement on hard tasks indicates that SF enables the model to accurately capture object locations and focus on their relative spatial relationships, rather than relying on shortcut correlations such as background or lighting cues.

## 3.3 COMPONENT-WISE ANALYSIS

We further investigate the effect of SF by varying multiple model components and training conditions. Below, we provide a detailed analysis of each ablation. We take OpenVLA-OFT (Kim et al., 2025) as the base model and conduct experiments on the LIBERO benchmark with a single H100 because of limitations of computational resources.

**Target Representation.** We examine whether the effectiveness stems primarily from our proposed paradigm or from the quality of the target representations, shown in Tab. 2. Both SigLIP (Zhai et al., 2023) and DINOv2 (Oquab et al., 2023) are widely adopted vision backbones pretrained on large-scale 2D image datasets. SigLIP excels at semantic understanding through robust image-text alignment, whereas DINOv2 offers stronger visual grounding owing to its fine-grained spatial representations. VGGT (Wang et al., 2025b), trained on 2D–3D paired datasets, possesses powerful spatial perception abilities for predicting 3D attributes. All models with SF alignment get higher success rates compared to the base model, which shows that visual embedding alignment serves as a general paradigm to implicitly enhance visual perception. Using VGGT as the target representation yields the highest success rates, demonstrating that compensating for the lack of 3D understanding is crucial for VLA models. Furthermore, we find that adding positional embedding (Ranftl et al., 2021) to the target representation significantly enhances performance on long-horizon tasks. The reason is that the aligned visual embeddings are utilized in an auto-regressive manner within VLA, where the relative position of tokens plays a critical role.

Table 2: **Component Analysis** on LIBERO benchmark. PE denotes positional embedding. **Bold** means the best performance. Experiments are conducted on 1×H100.

| Target Representation | Aligned Layer$^{th}$ | Training Iterations | Training Data | Spatial SR (%) | Object SR (%) | Goal SR (%) | Long SR (%) | Average SR (%) |
|---|---|---|---|---|---|---|---|---|
| ✗ | ✗ | 150K | 100% | 96.8 | 94.8 | 92.8 | 86.2 | 92.7 |
| SigLIP | 24 | 150K | 100% | 95.2 | 94.8 | 94.0 | 91.8 | 94.0 |
| DINOv2 | 24 | 150K | 100% | 93.4 | 95.2 | 93.8 | 93.8 | 94.1 |
| VGGT w/o PE | 24 | 150K | 100% | **97.8** | **100.0** | 96.6 | 84.4 | 94.7 |
| VGGT | 24 | 150K | 100% | 97.2 | 99.2 | **96.8** | **94.2** | **96.9** |
| VGGT | 1 | 150K | 100% | 96.8 | 99.4 | **99.0** | 83.0 | 94.6 |
| VGGT | 8 | 150K | 100% | 96.2 | 98.4 | 95.6 | 92.4 | 95.7 |
| VGGT | 16 | 150K | 100% | 97.4 | 98.8 | 95.8 | 83.2 | 93.8 |
| VGGT | 24 | 150K | 100% | 97.2 | 99.2 | 96.8 | **94.2** | **96.9** |
| VGGT | 32 | 150K | 100% | **98.8** | **99.4** | 96.2 | 84.8 | 94.8 |
| VGGT | 24 | 2K | 100% | 70.6 | 89.8 | 87.0 | 43.4 | 72.7 |
| VGGT | 24 | 5K | 100% | 93.8 | 94.8 | 94.6 | 66.6 | 87.5 |
| VGGT | 24 | 20K | 100% | 96.8 | 99.0 | 93.8 | 85.2 | 93.7 |
| VGGT | 24 | 50K | 100% | 97.0 | 99.0 | 96.2 | 93.6 | 96.5 |
| VGGT | 24 | 150K | 100% | **97.2** | **99.2** | **96.8** | **94.2** | **96.9** |
| VGGT | 24 | 150K | 1% | 32.8 | 67.8 | 44.8 | 23.6 | 42.3 |
| VGGT | 24 | 150K | 5% | 73.2 | 83.4 | 80.6 | 66.0 | 75.8 |
| VGGT | 24 | 150K | 100% | **97.2** | **99.2** | **96.8** | **94.2** | **96.9** |

**Alignment at Different VLA Layers.** We further investigate the effect of supervising different transformer layers of VLA, shown in Tab. 2. Our VLM backbone (Karamcheti et al., 2024) contains 32 causal attention layers. The results indicate that supervising relatively deep but not the deepest layers yields the most effective alignment. This is because supervising deep features implicitly enforces shallow features to be aligned more closely with spatial representations, thereby yielding improved spatial understanding at the global level. In contrast, constraining shallow features may cause the already aligned representations to gradually lose spatial information in subsequent layers. Besides, the visual and language modalities of VLA tend to converge into a modality-agnostic space with the layer increasing (Huang et al., 2024). Therefore, the last several layers lose more vision-specific features, making them less amenable to the supervision of target vision representation. This trade-off allows the 24th layer to achieve the best alignment performance.

**Training efficiency.** We analyze whether SF helps improve training efficiency, shown in Tab. 2 and Fig. 5(a). We report the task success rates vs. training iterations before and after representation

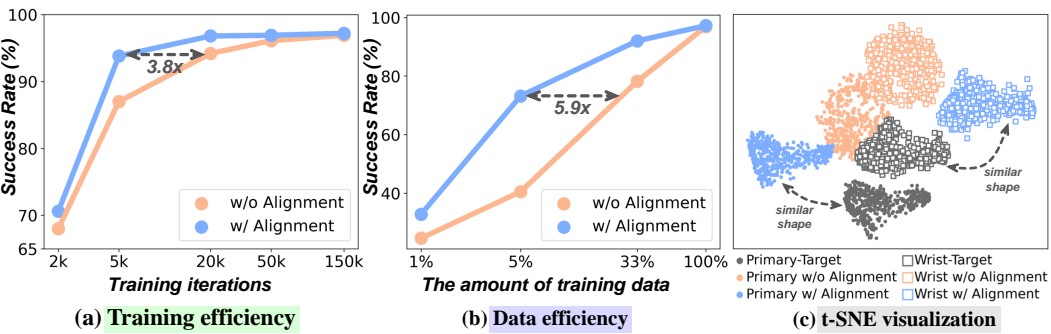

Figure 5: **(a)** We report the success rates vs. training iterations before and after representation alignment. **(b)** We report the success rate vs. training data before and after representation alignment. **(c)** The aligned representation exhibits almost the same distribution shape as the target.

alignment. The results illustrate that training with alignment significantly speeds up the convergence, achieving the same success rates 3.8× more quickly than the base model. We hypothesize that our SF serves as an efficient learning pathway, enabling VLA models to acquire visual understanding and rapidly capture essential spatial relationships.

**Data efficiency.** We analyze how SF helps improve data efficiency, shown in Tab. 2 and Fig. 5(b). We uniformly sample 1%, 5%, and 33% of the entire dataset to create mini datasets. To ensure fairness, we use the cosine-annealing rather than a multi-step training scheduler. SF reaches 75.8% success rates with only 5% data. It also achieves 25.8% higher success rates in terms of the same data amounts and reaches 5.9× more efficient in terms of the same success rates. Since the target representation is derived from pre-trained perception models, it can capture essential scene-general information. With the guidance of this target representation, the VLA can learn the core features with little conductive bias from only a small amount of robotic data, thereby improving task success rates. Given the scarcity of real-world data, this capability is particularly valuable for robotic applications.

**The t-SNE visualization.** We further visualize the degree of representation alignment. As shown in Fig. 5(c), it can be seen that after alignment, the VLA feature exhibits almost the same distribution shape as that of the target, while its cluster center remains independent from the target. This demonstrates that SF not only forces the VLA model to learn spatial comprehension ability but also ensures that its visual modality preserves original representational identity. More in-depth explanations can be seen in Appendix Sec. D.

# 4 REAL-WORLD EXPERIMENTS

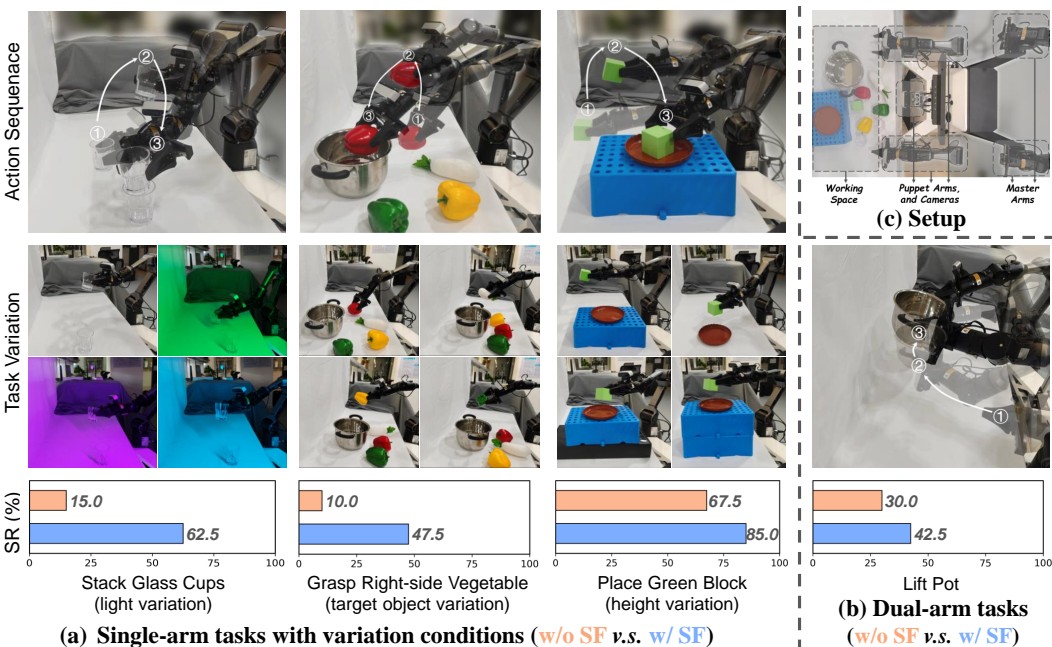

Figure 6: **Real-world Experiments. (a)** A set of single-arm tasks across various visual and spatial conditions. For each task, we train a unified model to face all variations and report the success rate. **(b)** Dual-arm tasks to measure the spatial horizontal balance ability. **(c)** Top-view robot setup.

We conduct real-world experiments to evaluate the effectiveness and data efficiency of our SF across highly variable environments.

## 4.1 EXPERIMENTAL SETUP

As shown in Fig. 6(c), we conduct real-robot experiments on the bimanual AgileX platform. Each arm consists of a 6-DoF Piper manipulator and a 1-DoF gripper. A primary camera and two wrist

cameras are installed on the body and the arms. We design a comprehensive set of tasks that covers various axes of spatial capabilities: *stack glass cups* with **light variation**, *grasp right-side vegetable* with **target object variation**, *place green block* with **height variation**, and *lift pot* with **new embodiments**. To examine the **data efficiency**, our model is trained on only 40 demonstrations for single-arm tasks and 20 for bimanual tasks. For evaluation, we test 10 trials per variation (40 trials in total) for each single-arm task, and 20 trials for the dual-arm task. We report the success rate (SR) of each task as the evaluation metric.

## 4.2 RESULT ANALYSIS

Fig. 6 shows that our SF achieves higher success rates across all tasks, showing considerable improvements in data efficiency. This ability is critical for real-world deployment. Specifically, in the *stack glass cups* task, transparent cups reflect varying surrounding lighting colors, making it highly deceptive. Our SF reaches 47.5% higher success rates than the base model because SF captures the underlying spatial relationships rather than overfitting the spurious correlations. In the *grasp right-side vegetable* task, different target objects require distinct gripper poses and clamping widths. Our performance demonstrates an understanding of the object's 3D appearance. For the *place green block* task, varying placement heights require precise estimation of spatial height information. Benefit from spatial feature learning, SF achieves an 85% success rate. Results of the bimanual *lift pot* task indicate the adaptability in the new configuration as well as spatial awareness of the pot's horizontal balance to prevent tilting. The performances across all tasks in Fig. 6(a) and (b) demonstrate that SF substantially improves task success rates, indicating its strong spatial comprehension and data utilization capabilities.

## 5 RELATED WORK

**Vision-language-action Models.** Given natural language instructions and scene observations, VLA aims to produce executable actions for robots. Prior works leverage vision and language foundation models to enhance robotic capabilities for low-level object localization (Stone et al., 2023; Gadre et al., 2023) as well as high-level reasoning and planning (Brohan et al., 2022; Zhao et al., 2023; Huang et al., 2023). With the development of VLMs (Liu et al., 2024a; Karamcheti et al., 2024; Beyer et al., 2024), numerous VLA research (Zitkovich et al., 2023; Deng et al., 2025; Li et al., 2024a; Song et al., 2025a; Li et al., 2024b) utilize VLMs with action experts to learn the generalizable action and language knowledge. OpenVLA (Kim et al., 2024) proposes the first open-source VLA model pretrained on large-scale robotic datasets (O'Neill et al., 2024b). However, these methods primarily focus on 2D image information and lack an accurate comprehension of the 3D physical world. Recent studies enhance the spatial perception of VLAs through depth estimation (Zhen et al., 2024; Qu et al., 2025; Zhang et al., 2025), point cloud injection (Li et al., 2025a; Sun et al., 2025), and spatial projection (Li et al., 2025b; Argus et al., 2025). Nevertheless, these approaches exclusively enrich the visual inputs of VLAs with spatial information, while overlooking that the visual embeddings as intermediate scene representations also play a crucial role in generating action tokens. In contrast, we argue that aligning the visual embeddings of VLAs with external powerful 3D representations (Wang et al., 2025b) is an efficient supervision to guide VLAs with spatial learning.

**Representation Supervision.** The latent representations of models can be supervised through reconstruction or alignment to facilitate downstream task adaptation. As for reconstruction-based supervision, L-DAE (Chen et al., 2024) and Genhancer (Ma et al., 2025) enable generative models to enhance visual representations. ROSS (Wang et al., 2024; 2025a) supervises the visual embeddings of VLMs for VQA tasks by employing a denoising architecture (Ho et al., 2020) to reconstruct input images. ReconVLA (Song et al., 2025c) adopts reconstructions on the gazing area as a supervised objective to guide VLAs in allocating attention to target objects. However, the reconstruction supervision may not be suitable for VLAs to learn effective representations, as it fails to filter out redundant details (LeCun, 2022; Assran et al., 2023). As for alignment-based supervision, (Wu et al., 2025) and REPA (Yu et al., 2024) directly align the intermediate hidden states of image and video generation models with external pretrained visual encoders (Oquab et al., 2023; Wang et al., 2025b). 3DRS (Huang et al., 2025) introduces 3D representation supervision to strengthen its spatial grounding capability. The cosine similarity is commonly used as a loss function to aligns hidden states at the pixel-patch level independently. Our work also shares some similarities, where we follow the

alignment-based paradigm to guide the intermediate representations of models towards geometry-aware structures. Specifically, owing to the global attention across patches within the external 3D foundation model (*e.g.* VGGT), this alignment yields stable 3D representations for the whole scene.

## 6 CONCLUSION

In this paper, we investigated how to implicitly force VLAs to develop 3D perception and comprehension capabilities. We begin with a lightweight depth probing experiment to investigate the insufficiency of spatial reasoning in current VLA models. Consequently, we propose **Spatial Forcing** (SF), a simple yet effective method that aligns the visual embeddings in VLAs with external spatial representations extracted from 3D foundation models. Finally, the simulation experiments prove that SF can enhance performance, accelerate training speeds, and improve data efficiency. The real-world experiments prove its spatial comprehension capabilities across diverse robotic tasks.

## ACKNOWLEDGMENTS

This work was supported in part by the Natural Science Foundation of China under Grant 62403401, in part by the Guangdong Basic and Applied Basic Research Foundation under Grant 2024A1515011992, in part by the Guangdong Provincial Project under Grant 2024QN11X127, in part by the Shenzhen Science and Technology Program under Grant CJGJZD20240729141702003, and in part by the AI Research and Learning Base of Urban Culture under Grant 2023WZJD008.

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

## A  THE USE OF LARGE LANGUAGE MODELS (LLMS)

In preparing this paper, large language models (LLMs) were utilized solely as general-purpose writing aids for light proofreading and language polishing. This assistance was limited to minor improvements in grammar, phrasing, and style. All core intellectual contributions, including the research ideas, problem formulation, methodology, experimental design, data analysis, code development, and substantive writing, were conceived and executed by the authors. No LLM was used to generate original content, design experiments, analyze data, write code, or draft any part of the manuscript. The authors thoroughly reviewed and verified all text, and they take full responsibility for the entire content of the paper. The LLM should not be considered an author or contributor.

## B  REPRODUCIBILITY STATEMENT

We have taken several measures to ensure the reproducibility of our work. The main paper provides detailed descriptions of the model architecture, training objectives, evaluation protocols, and implementation details. Additional hyper-parameter settings and real-world data collection settings are presented in the appendix. Furthermore, we include a demo source code package in the supplementary materials to facilitate the reproduction of our results.

Table 3: weight factor of alignment loss.

| $\alpha$ | 0 | 0.02 | 0.1 | 0.5 | 2.5 | 12.5 |
|---|---|---|---|---|---|---|
| SR (%) | 73.2 | 92.2 | 92.8 | **93.6** | 86.6 | 81.2 |

## C  WEIGHT FACTOR

The hyper-parameter $\alpha$ is used to control the relative weight of the alignment loss in Eq. (4). Appropriate weight can implicitly force the model to develop spatial comprehension capability. However, the excessively large weight may destabilize the VLA visual modality and interfere with the original robot action prediction. As shown in Tab. 3, the model performs best under the $\alpha = 0.5$, which is the default setting in all other experiments.

## D  EXPLANATIONS OF t-SNE RESULTS

This section provides a more detailed explanation of the t-SNE visualization used in Sec. 3.3 to demonstrate the effectiveness of our SF. T-Distributed Stochastic Neighbor Embedding (t-SNE) is a non-linear dimensionality reduction technique primarily used for visualizing high-dimensional features into a low-dimensional space (typically 2D or 3D). It works by modeling the similarity between high-dimensional data points as a conditional probability. Specifically, it computes the probability that a point $x_i$ would pick another point $x_j$ as a neighbor, based on the Gaussian distribution centered on $x_i$.

The **distribution shape** of a t-SNE cluster reveals the internal relative similarities and discrepancies among the features within that set. A similar distribution shape indicates that the relational structure between features has become isomorphic. In our results, the VLA feature with SF alignment exhibits almost the same distribution shape as that of the target. This means that, by forcing the VLA model's features to adopt a similar relational geometry, our SF method is doing more than just a simple linear mapping function. It is forcing the VLA model to learn the underlying manifold of the target's spatial representation. The VLA model learns not just where individual features should be, but how the entire feature space is structured with respect to spatial concepts.

The **center** of a t-SNE cluster is roughly the densest, most typical region of that feature set. The independence between two cluster centers indicates that their local structures are preserved separately. In our results, the center of aligned VLA features remains independent from that of the target, which signifies that the alignment process has not caused a representational collapse. If the alignment were merely forcing the VLA features to exactly replicate the target features, the two clusters would overlap entirely. The distinct cluster centers demonstrate that while the VLA features have adopted the relational structure of the target, they have done so without discarding the information that is unique to their own modality.

## E  REAL-WORLD DATA COLLECTION

During data collection, we use the master arms in a teleoperated manner to guide the puppet arms to finish tasks. Camera images and absolute joint angles are recorded as task-specific datasets at 30Hz. During training and inference, the model is fine-tuned separately for each task to control the puppet arms in task execution.

## F  DETAILS OF COMPARED MODELS

$\pi_0$ **(Black et al., 2024)**   $\pi_0$ is a vision-language-action model for general robot control that integrates a pre-trained vision-language model (VLM) with a novel flow matching action expert. This architecture enables the model to produce continuous, high-frequency actions. The model is trained with a two-stage recipe: broad pre-training on a large-scale, diverse, cross-embodiment dataset, followed by optional fine-tuning on high-quality data. In evaluations, $\pi_0$ excels at following language commands and significantly outperforms methods designed specifically for dexterous manipulation

tasks. Furthermore, the model can be adapted to master exceptionally complex, multi-stage tasks that take 5 to 20 minutes to complete, such as folding laundry and bussing cluttered tables.

**Openvla-OFT (Kim et al., 2025)** Openvla-OFT introduces an Optimized Fine-Tuning (OFT) recipe designed to enhance both performance and inference efficiency of VLAs when adapting them to specific robotic tasks. The recipe integrates parallel decoding, action chunking, a continuous action representation, and a simple L1 regression objective to improve inference efficiency. For tasks requiring precise language understanding, the recipe is further augmented with FiLM (Feature-wise Linear Modulation) (Perez et al., 2018) to strengthen language grounding. On the LIBERO simulation benchmark (Liu et al., 2023a), OpenVLA-OFT boosts the success rate to 97.1% while increasing action generation throughput by 26x. In real-world evaluations on a bimanual ALOHA robot (Zhao et al., 2024), it outperforms strong fine-tuned VLAs like $\pi_0$ (Black et al., 2024) and RDT-1B (Liu et al., 2024b), as well as policies trained from scratch, by up to 15% (absolute) in average success rate on dexterous tasks.

**Diffusion Policy (Chi et al., 2023)** Diffusion Policy is a visuomotor policy that represents the robot's behavior as a conditional denoising process. Instead of directly predicting an action, Diffusion Policy learns the gradient of the action distribution and iteratively refines a randomly sampled action through a series of denoising steps with respect to the gradient field. This formulation enables the model to gracefully handle multimodal action distributions and high-dimensional action spaces, leading to impressive training stability. Across 15 different tasks from 4 different robot manipulation benchmarks (Florence et al., 2019; Gupta et al., 2019; Mandlekar et al., 2021; Shafiullah et al., 2022), Diffusion Policy consistently outperforms existing state-of-the-art robot learning methods with an average improvement of 46.9%.

**TraceVLA (Zheng et al., 2025)** TraceVLA provides a visual trace prompting technique that enhances the spatial-temporal awareness of generalist robotic policies. The method employs an off-the-shelf point tracker Co-Tracker (Karaev et al., 2024) to generate trajectories of the robot's past movements, which are then visually overlaid onto the current observation as an additional input prompt for the VLA Model. Evaluations show that TraceVLA outperforms the OpenVLA (Kim et al., 2024) baseline by 10% in the SimplerEnv simulation and by 3.5x on real-world robot tasks, showcasing robust generalization across diverse embodiments and scenarios.

**Octo (Ghosh et al., 2024)** Octo is an open-source, transformer-based model for robotic manipulation pretrained on 800k trajectories from the Open X-Embodiment (OXE) dataset (O'Neill et al., 2024a). The architecture pairs a large transformer backbone for processing multimodal inputs with a lightweight diffusion head that generates expressive, continuous actions. The model can be instructed via language commands or goal images and demonstrates strong zero-shot performance, outperforming RT-1-X (O'Neill et al., 2024a) by 29% on average. Its compositional design makes it a versatile initialization for data-efficient finetuning. On average, finetuned Octo policies outperform training from scratch by 52%, successfully adapting to novel sensors, new action spaces, and entirely new embodiments. Crucially, this adaptation is achieved with only 100 demonstrations and a few hours of training on a single consumer GPU.

**Openvla (Kim et al., 2024)** OpenVLA is a 7B-parameter, open-source Vision-Language-Action (VLA) model designed for generalist robotic manipulation. The model's architecture is built on a Llama 2 backbone (Touvron et al., 2023) combined with a powerful visual encoder that fuses features from both DINOv2 (Oquab et al., 2023) and SigLIP (Zhai et al., 2023), enabling strong spatial reasoning and semantic understanding. Pretrained on a diverse collection of 970k real-world robot demonstrations from the Open X-Embodiment dataset (O'Neill et al., 2024a), OpenVLA outperforms the closed-source 55B RT-2-X model (O'Neill et al., 2024a) by 16.5% in absolute success rate across multiple robots and 29 tasks on the WidowX (BridgeData V2) (Walke et al., 2023) and Google Robot platforms. OpenVLA also introduces robust and efficient fine-tuning strategies, such as LoRA (Hu et al., 2022), which allow the model to be quickly adapted to new tasks and robots on consumer-grade GPUs.

**Dita (Hou et al., 2025)** Dita is a scalable framework for generalist robotic learning that leverages a Diffusion Transformer (DiT) (Peebles & Xie, 2023) to directly denoise continuous action sequences.

The architecture features an "in-context conditioning" mechanism where a causal transformer directly processes raw visual tokens from historical observations to inform the denoising of future actions, enabling fine-grained alignment and explicit modeling of environmental nuances. Pretrained on the large-scale OXE dataset, the lightweight 334M Dita model achieves state-of-the-art or competitive performance across extensive simulation benchmarks like LIBERO, CALVIN (Mees et al., 2022), and ManiSkill2 (Gu et al., 2023). Furthermore, it demonstrates robust real-world adaptation, successfully executing complex, long-horizon manipulation tasks with just 10-shot finetuning.

**CoT-VLA (Zhao et al., 2025)**   CoT-VLA is a VLA model that incorporates visual chain-of-thought (CoT) reasoning, where it first auto-regressively generates a future subgoal image as an intermediate reasoning step before predicting the action sequence. The model is built upon the VILA-U multimodal foundation model (Wu et al., 2024) and features a hybrid attention mechanism, using causal attention for subgoal image generation and full attention for multi-step action prediction. The training dataset includes not only robot demonstrations from the Open X-Embodiment dataset but also action-less video datasets like EPIC-KITCHENS (Kapidis et al., 2019), allowing the model to improve visual reasoning from unlabeled sources. In evaluations, CoT-VLA achieves a 6% average improvement over OpenVLA on the LIBERO simulation benchmark and a 17% improvement in real-world manipulation tasks.

**$\pi_0$-FAST (Pertsch et al., 2025)**   FAST introduces an efficient action tokenization method for VLA models that enables the training of auto-regressive policies on high-frequency, dexterous manipulation tasks where previous binning schemes failed. The approach, named Frequency-space Action Sequence Tokenization (FAST), leverages the discrete cosine transform (DCT) to convert action trajectories into the frequency domain, where the signal's core information is naturally concentrated into a few low-frequency coefficients. This method effectively compresses actions into a compact set of discrete tokens, significantly reducing redundancy. When integrated with the $\pi_0$ backbone, the resulting auto-regressive policy ($\pi_0$-FAST) matches the performance of the original $\pi_0$ model on complex, long-horizon tasks like laundry folding, while reducing training time by up to 5x.

**UniVLA (Bu et al., 2025)**   UniVLA introduces a framework for generalist robotic policies that learns a unified, task-centric latent action space from videos, uniquely enabling it to leverage diverse data sources (including human videos) without explicit action labels. The core of the method is an unsupervised latent action model that uses a VQ-VAE to discretize task-relevant dynamics from paired video frames within the DINOv2 feature space. These quantized latent actions then serve as pseudo-labels to pretrain an auto-regressive vision-language policy. In evaluations, UniVLA significantly outperforms OpenVLA, achieving a 95.2% success rate on the LIBERO benchmark (an 18.7% absolute improvement) and a 36.7% absolute improvement in real-world deployment tasks, while using less than 1/20 of the pretraining compute.

**SpatialVLA (Qu et al., 2025)**   SpatialVLA is a spatial-enhanced VLA model designed to improve 3D spatial understanding. The architecture introduces two key innovations: Ego3D Position Encoding, which injects 3D spatial context from depth information into the input observation of a PaliGemma 2 backbone (Steiner et al., 2024), and Adaptive Action Grids, a novel action representation that discretizes continuous robot movements into adaptive spatial grids based on the data distribution. Pretrained on 1.1 million real-world robot episodes, SpatialVLA demonstrates superior zero-shot performance and efficient adaptation capabilities. In extensive evaluations across 24 real-world tasks and 3 simulation environments (Li et al., 2024c; Liu et al., 2023a; Walke et al., 2023), it achieves state-of-the-art results, significantly outperforming models like OpenVLA and RoboVLM (Li et al., 2024b), particularly in tasks requiring precise spatial reasoning and generalization to new robot setups.

**GeoVLA (Sun et al., 2025)**   GeoVLA is a VLA framework designed to improve robotic manipulation by explicitly integrating 3D geometric information alongside standard 2D visual inputs. Its novel dual-path architecture features a standard VLM for processing 2D vision and language, together with a custom Point Embedding Network (PEN) that extracts geometric features from point clouds derived from depth maps. These multimodal embeddings are then fused by a 3D-enhanced Action Expert (3DAE) to generate precise and continuous actions. In evaluations, GeoVLA achieves state-of-the-art performance, outperforming OpenVLA-OFT on the LIBERO benchmark and Dita

on ManiSkill2 (Gu et al., 2023). In real-world experiments, it demonstrates superior robustness to 3D variations such as changes in object height, scale, and camera viewpoint, outperforming strong baselines like $\pi_0$ by 28.8% in average success rate.

**3D-CAVLA (Bhat et al., 2025b)**  3D-CAVLA introduces a framework that enhances Vision-Language-Action models with improved 3D spatial awareness and reasoning to boost generalization on unseen tasks. Built upon OpenVLA-OFT, the model integrates three key modifications: chain-of-thought-style narrative prompts to enrich task context, 3D features derived from point clouds to improve depth perception, and task-oriented region-of-interest pooling to focus visual attention. The model achieves a near-perfect 98.1% average success rate on standard LIBERO tasks. Furthermore, it demonstrates a significant 8.8% absolute improvement over OpenVLA-OFT on a newly proposed benchmark of 10 zero-shot tasks derived from the LIBERO environment, highlighting its superior generalization capabilities.

## G  ADAPTIVELY CHOOSING ALIGNMENT DEPTH LAYER

We develop an adaptive strategy to automatically choose the alignment depth layer of VLAs, and we find this strategy is effective. Specifically, following the core idea of MOE soft gating mechanism (Li et al., 2019; Shazeer et al., 2017), firstly we employ a trainable weight matrix $M_g \in \mathbb{R}^{L \times D}$, where $L$ is the total amount of VLA layers (*e.g.*, 32 for LLaVA), $D$ is the feature dimension. Then, pool the VLAs' visual tokens along the token axis:

$$P = \frac{1}{N} \sum_{t=1}^{N} x_t^{\mathcal{V}}, \tag{5}$$

where $\{x_t^{\mathcal{V}}\}_{t=1}^{N}$ is N visual token of VLAs. If the visual tokens $^l x^{\mathcal{V}}$ are gained at the $l_{th}$ layer of VLAs, we denote the corresponding pooling feature as $P(l)$, where $l = \{1, 2, ..., L\}$.

Next, calculate the gating score scalar for each layer:

$$s_l = M_g(l)P(l)^{\mathrm{T}}. \tag{6}$$

Finally, we use the mixtual visual feature to calculate the align loss:

$$\mathcal{F}_{align}[\sum_{l=1}^{L} \frac{exp(s_l)}{\sum_{j=1}^{L} exp(s_j)} {}^l x^{\mathcal{V}}, f^{3D}]. \tag{7}$$

We report the updated success rate on LIBERO. As shown in the Tab. 4 below, the adaptive layer selection strategy shows clear improvements. And we provide the gating score distribution histograms across layers in Fig. 7.

Table 4: The Ablation of Supervising Different Transformer Layers of VLA.

| Aligned Layer[th] | Spatial | Object | Goal | Long | Average |
|---|---|---|---|---|---|
| 1 | 96.8 | **99.4** | **99.0** | 83.0 | 94.6 |
| 8 | 96.2 | 98.4 | 95.6 | 92.4 | 95.7 |
| 16 | 97.4 | 98.8 | 95.8 | 83.2 | 93.8 |
| 24 | 97.2 | 99.2 | 96.8 | 94.2 | 96.9 |
| 32 | **98.8** | **99.4** | 96.2 | 84.8 | 94.8 |
| adaptive | 98.6 | **99.4** | 98.8 | **95.4** | **98.1** |

## H  ADDATIONAL REAL-WORLD SETTINGS

### H.1  COMPLEX COLOR, CLUTTER, INSTRUCTION CHALLENGES

As shown in Fig. 8, the target objects include red bell pepper, green bell pepper, red carrot, or green cube. This experimental setting is provided in the revised appendix. Models are trained on 40 demonstrations and evaluated over 40 trials. For each evaluation trial, unseen objects appear on the table and are randomly positioned. The results are shown in Tab. 5.

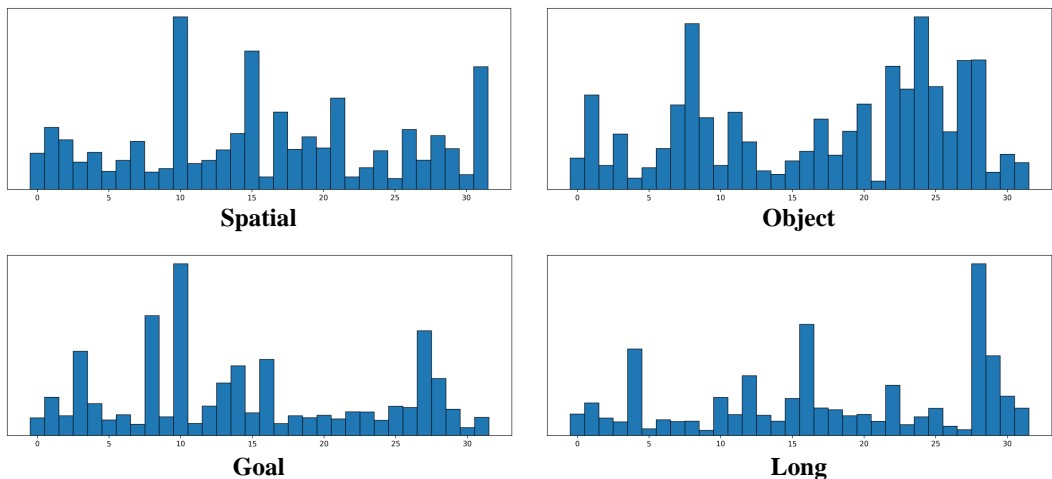

Figure 7: The MOE gating score distribution histograms on the LIBERO benchmark of adaptive layer selection strategy.

Table 5: Real-world results with complex color, unseen clutter, diverse instruction challenges.

| Method | Success Rate |
|--------|--------------|
| w/o SF | 22.5 |
| w/ SF | 37.5 |

## H.2 COMPLEX SPATIAL CHALLENGES

As shown in Fig. 9, "Transfer Block from Cardboard" task requires the robot to transfer a block from one cardboard box to another. For this task, the highly constrained spatial positions require precise perception of the box openings and object locations. The results are shown in Tab. 6.

Table 6: The real-world results on the *Transfer Block from Cardboard* task.

| Method | Success Rate |
|--------|--------------|
| OpenVLA-OFT | 12.5 |
| 3D-CAVLA | 27.5 |
| SF (ours) | 32.5 |

## I RECONSTRUCTION PROBING

To further validate that the aligned VLA visual tokens still contain non-spatial information (e.g., color), we extend the depth probing experiment in our paper to a color-reconstruction probing experiment. Specifically, we freeze the whole VLA model and train a two-layer MLP and a DPT head to simultaneously predict patch-level color and depth. The probing visualizations are provided in Fig. 10, which further confirms the rich color information in the VLA visual tokens after the alignment.

## J DOWNSTREAM TASK SUPERVISION STRATEGY

To understand what specific geometric properties are most dominant in these features, we design a paradigm to directly align VLA with downstream properties of VGGT. Following the official implementation of VGGT, we mainly choose three geometric properties: depth, point maps, and dynamic tracks. Firstly, we utilize the task-specific heads of VGGT as VLA downstream heads.

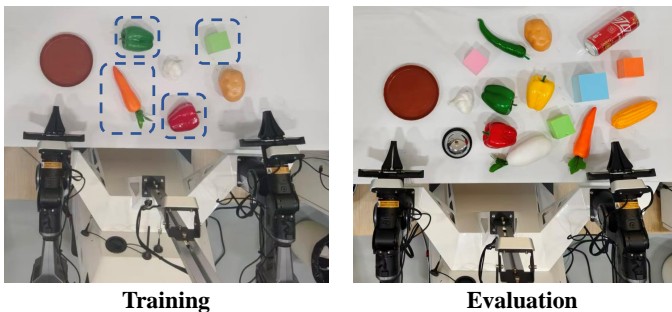

**Training**                    **Evaluation**

Figure 8: Additional real-world experiments with complex color, unseen clutter, diverse instruction challenges.

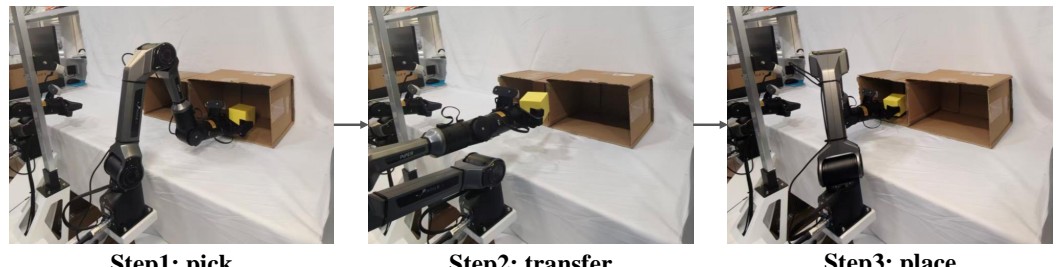

**Step1: pick**          **Step2: transfer**          **Step3: place**

Figure 9: Additional ablation experiments for more complex spatial tasks.

Then, the visual tokens of VLA are fed into the trainable VLA downstream heads to produce task-specific outputs (i.e., depth, point maps, or dynamic tracks). Finally, we treat the VGGT task-specific outputs as ground-truth and calculate the loss between these two outputs. The overall pipeline is shown in the Fig. 11.

## K  SCALING PERFORMANCE EVALUATION

To better evaluate our scaling performance on increasing data and training steps, we choose a difficult task in RoboTwin and collect additional data. The experiments are on the RoboTwin "beat block" task. The original data includes 50 episodes and we additionally collected 150 episodes, total of 200 episodes. The line charts of success rate comparison are shown in Fig. 12.

The x-axis of the original data-scaling curves is displayed in a logarithmic scale. Under the log scale, the curves visually appear much steeper. Therefore, as shown in Fig. 13, we provide the data-scaling curves using the linear scale x-axis, which reflects the absolute dataset size without bias. In addition, to make sure that these data-scaling curves fully converged, we conducted further experiments under the larger dataset sizes. Experimental results demonstrate that as the dataset size increases, the performance gains gradually become marginal for both the baseline and our SF method. However, we could find that under every setting of data sizes, our SF consistently outperforms the baseline. In particular, even when the data-scaling curves nearly converge under large data sizes, our SF still maintains a significantly higher success rate. This indicates that our SF remains effective even under large data scales.

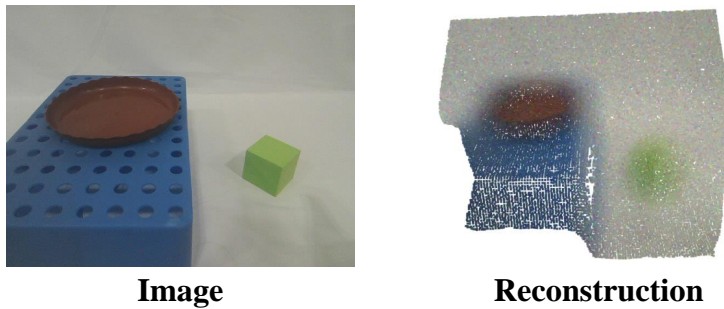

**Image**                    **Reconstruction**

Figure 10: Reconstruction probing results.

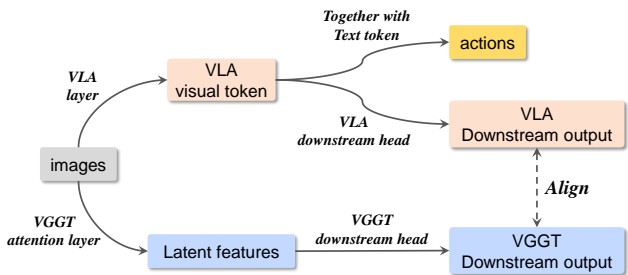

Figure 11: The pipeline of the downstream task supervision strategy.

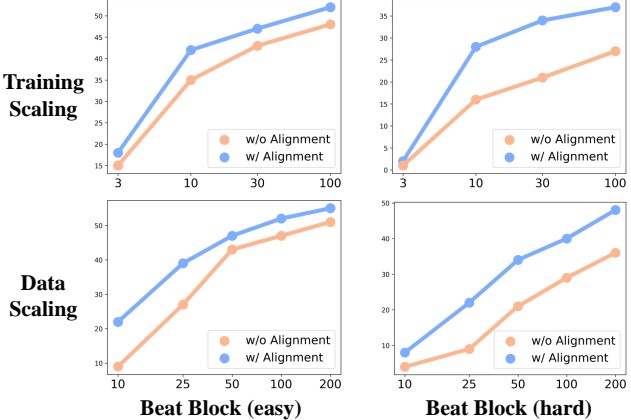

Figure 12: Additional ablation experiments for training iteration and data with the logarithmic scale x-axis.

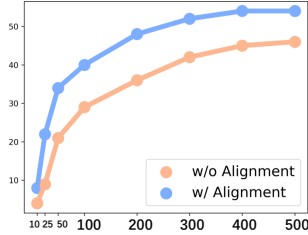

Figure 13: Data-scaling curves with the linear scale x-axis.

