# OpenReview forum: "Spatial Forcing: Implicit Spatial Representation Alignment for Vision-language-action Model"
_ICLR.cc/2026/Conference — ICLR 2026 Poster_

### Official Review · Reviewer_UKHb · 2025-10-20

**Soundness:** 2
**Presentation:** 3
**Contribution:** 3
**Rating:** 6
**Confidence:** 4

**Summary:**

To address the limitations of VLA models largely relying on 2D visual features and the difficulty of acquiring and representing explicit 3D inputs, this paper proposes Spatial Forcing (SF), a method that aligns the deep features of VLA models with target features embedded with spatial awareness, thereby enhancing the spatial reasoning capability of VLA models during task execution and improving action prediction accuracy. The target features containing 3D spatial information are provided by the VGGT model.

The authors investigate the importance of 3D spatial perception for the success rate of VLA tasks and validate the effectiveness of the SF method in enhancing 2D-based VLA models through multiple simulators and real-world experiments. Additionally, various visualization techniques demonstrate that the VLA model’s features are focused on spatial information and object geometry representation.

**Strengths:**

1.	Analysis of 3D spatial perception limitations in existing VLA methods: The paper identifies that most current VLA approaches lack understanding of 3D spatial information and highlights the limitations of existing attempts to incorporate 3D cues into VLA models.
2.	Deep insights into LLM feature layers: The authors investigate the features from different layers of the LLM and determine appropriate layers for optimal feature alignment.
3.	Efficient feature alignment with SF: SF is a training-only feature alignment method that significantly improves data efficiency and model capability, particularly in real-world scenarios.

**Weaknesses:**

1.	While SF demonstrates impressive capability in obtaining spatially informed representations, aligning VGGT features with deep layers of the LLM (e.g., the 24th layer) might potentially compromise the 2D image-derived features acquired in the preceding layers, such as semantic understanding and generalization ability. The authors could provide experimental verification, for example, through real-world experiments testing scene generalization or tasks requiring strong semantic comprehension.
2.	In the real-world experiments, there is no baseline model that explicitly takes 3D inputs. Given that SF emphasizes comparison with 3D-aware models in both method and simulation experiments, including an explicit 3D-input baseline in more spatially complex real-world scenarios could more convincingly demonstrate the advantages of SF.

**Questions:**

In Table 2, the first row represents a model without target representation and without feature alignment, which uses OpenVLA-OFT as the base model. However, the reported performance in Table 2 does not match the results in Table 1, which may affect the validity of the ablation study. It would be helpful if the authors could explain the cause of this discrepancy or provide additional experimental details.

---

> ### Author Response · Authors · 2025-11-22
> **Response to Reviewer UKHb (1/2)**
>
> We deeply appreciate your insightful efforts and suggestions in reviewing our manuscript. We respond to each of your comments one by one in what follows. In the revised manuscripts, we mark our major revisions as “blue”.
> ### **[W1] While SF demonstrates impressive capability in obtaining spatially informed representations, aligning VGGT features with deep layers of the LLM (e.g., the 24th layer) might potentially compromise the 2D image-derived features acquired in the preceding layers, such as semantic understanding and generalization ability. The authors could provide experimental verification, for example, through real-world experiments testing scene generalization or tasks requiring strong semantic comprehension.**
>
> **Reply:**
> **Scene Generalization.** Firstly, the real-world example in our main manuscript has already demonstrated that our model achieves high success rates when faced with variations in lighting, target objects, and table height, indicating that the aligned VLA visual features have generalization to scene variations.
>
> Secondly, to verify generalization under random initial pose distribution, we additionally evaluate three single-arm tasks with variation. As shown in the table below, the success rate (%) results demonstrate that our approach still exhibits improved spatial capabilities in out-of-distribution (OOD) initial pose distribution scenarios, further confirming that the aligned VLA visual features remain generalization to large-scale viewpoint variations.
>
> \begin{array}{c|ccc} \hline  \text{Method} & \text{Stack Glass Cups} & \text{Grasp Right-side Vegetable} & \text{Place Green Block}\newline \hline \text{w/o SF} & 5 & 5 & 22.5 \newline \text{w/ SF} & 27.5 & 17.5 & 40 \newline \hline \end{array}
>
> **Tasks Requiring Semantic Comprehension.** To verify generalization when facing unseen objects and unseen prompts, we additionally design a real-robot manipulation task: grasping different objects among highly cluttered objects on a table. The target objects include red bell pepper, green bell pepper, red carrot, or green cube. The experimental setting is provided in the revised Appendix H1. For each evaluation trial, all objects are randomly positioned, and language instructions are diversified using GPT-4o. Since many distractor objects share visual features with the target objects, understanding the semantic meaning of the target objects is critical for the task success. The success rate (%) table below demonstrates that SF preserves the semantic comprehension of the VLA model.
>
> \begin{array}{c|c} \hline
> \text{Method} & \text{Success Rate}
> \newline \hline
> \text{w/o SF} & 22.5
> \newline
> \text{w/ SF} & 37.5
> \newline \hline \end{array}
>
> ### **[W2] In the real-world experiments, there is no baseline model that explicitly takes 3D inputs. Given that SF emphasizes comparison with 3D-aware models in both method and simulation experiments, including an explicit 3D-input baseline in more spatially complex real-world scenarios could more convincingly demonstrate the advantages of SF.**
>
> **Reply:**
> Following your instructions, we choose an explicit 3D-input baseline, design new spatially complex tasks, and conduct real-world experiments to more convincingly demonstrate the advantages of our SF.
>
> **Explicit 3D-input Baseline.** We use the open-source 3D-CAVLA$^{[1]}$ that explicitly utilize 3D depth as the baseline for comparison.
>
> **Spatially Complex Tasks.** Firstly, we follow our real-world protocol to evaluate it on our existing "Place Green Block" task. For this task, varying placement heights require precise estimation of spatial height information. In addition, we design a new real-world task: "Transfer Block From Cardboard", which requires the robot to transfer a block from one cardboard box to another. For this task, the highly constrained spatial positions require precise perception of the box openings and object locations. The detailed experimental setting is provided in the revised Appendix H2.
>
> **Results.** As shown in the table below, both 3D methods achieve higher success rates (%) than 2D method (OpenVLA-OFT). Compared with the explicit 3D-input method (3D-CAVLA), our SF exhibits superior performance on both tasks, demonstrating the spatial understanding and precision action of our methods. The results further prove the effectiveness of our implicit 3D learning paradigm.
>
> \begin{array}{c|cc} \hline
> \text{Method} & \text{Place Green Block} & \text{Transfer Block From Cardboard}
> \newline \hline
> \text{OpenVLA-OFT} & 67.5 & 12.5
> \newline
> \text{3D-CAVLA} & 75.0 & 27.5
> \newline
> \text{SF (ours)} & 85.0 & 32.5
> \newline \hline \end{array}
>
> [1] 3D CAVLA: Leveraging Depth and 3D Context to Generalize Vision Language Action Models for Unseen Tasks, arxiv 2025.

---

> ### Author Response · Authors · 2025-11-22
> **Response to Reviewer UKHb (2/2)**
>
> ### **[Q1] In Table 2, the first row represents a model without target representation and without feature alignment, which uses OpenVLA-OFT as the base model. However, the reported performance in Table 2 does not match the results in Table 1, which may affect the validity of the ablation study. It would be helpful if the authors could explain the cause of this discrepancy or provide additional experimental details.**
>
> **Reply:**
> The discrepancy is caused by the difference in the number of GPUs for training. The main results in Table 1 are conducted on 8 $\times$ H100 GPUs, while the ablation experiments in Table 2 are conducted on a single H100 GPU because of limitations of computational resources. Therefore, both the base model and SF have discrepancies in the success rate between these two tables.

---

### Official Review · Reviewer_giXr · 2025-10-25

**Soundness:** 3
**Presentation:** 3
**Contribution:** 3
**Rating:** 6
**Confidence:** 3

**Summary:**

This submission employs intermediate-layer alignment to enhance spatial comprehension.Rather than utilizing explicit 3D inputs or a depth predictor in the VLA pipeline, this method only relies on VGGT during the training phase.Evaluations in both simulated and real-world environments demonstrate that the proposed method outperforms existing 3D VLAs.

**Strengths:**

Other 3D-aware VLA methods explicitly feed depth maps into the VLA model, whereas the proposed method does not require the input of depth maps during the inference phase—thus enabling faster training and inference. The method is simple and straightforward, and evaluations indeed show better performance compared to existing counterparts.

**Weaknesses:**

As Figure 5 illustrates, while the method with alignment improves the success rate more rapidly than the method without alignment, as the data volume and training iterations increase, the two methods achieve the same performance. This indicates that the proposed method does not remain valid as the data size grows. This raises a question: if the data volume or training steps continue to increase, could the method without alignment exhibit stronger performance instead?

**Questions:**

The main question concerns the effectiveness of the alignment. Given that data size and computational power will undoubtedly continue to grow in the future, will the proposed method still hold its effectiveness at that point?

---

> ### Author Response · Authors · 2025-11-22
> **Response to Reviewer giXr**
>
> Thank you for your insightful questions and efforts in reviewing our manuscript. If we understand correctly, the mentioned weakness and questions are highly relevant. In the following, we will carefully respond to it point by point. In the revised manuscripts, we mark our major revisions as “blue”.
> ### **[W1 & Q1] As Figure 5 illustrates, while the method with alignment improves the success rate more rapidly than the method without alignment, as the data volume and training iterations increase, the two methods achieve the same performance. This indicates that the proposed method does not remain valid as the data size grows. This raises a question: if the data volume or training steps continue to increase, could the method without alignment exhibit stronger performance instead?**
> ### **The main question concerns the effectiveness of the alignment. Given that data size and computational power will undoubtedly continue to grow in the future, will the proposed method still hold its effectiveness at that point?**
>
> **Reply:**
> **Explanation for Figure 5.** Figure 5 in the main manuscript illustrates the improvement in training and data efficiency brought by SF on the LIBERO benchmark. It is worth noting that both the baseline and SF achieve nearly 100% success rates under full training iterations and data. This is because the LIBERO experimental setting is relatively simple and fixed, making high success rates easy to achieve. As the success rate is nearly saturated, it can not evaluate the scaling performance of our strategy.
>
> **Additional Experiments on scaling data and training iterations.** **Setup.** To better evaluate our scaling performance on increasing data and training steps, we choose a difficult task in RoboTwin and collect additional data. The experiments are on the RoboTwin "beat block" task. The original data includes 50 episodes and we additionally collect 150 episodes, total of 200 episodes. **Results.** The success rate (%) comparison is shown in the table below, and line charts for better visualization of the trends are provided in the revised Appendix K. The experimental results show that with the increasing training iterations and data amounts, our proposed SF continuously improves the performance and reaches higher success rates than the baseline, especially in more complex scenarios. This demonstrates the scaling performance of our SF with scaling data and training iterations.
>
> \begin{array}{cc|cc} \hline
> \text{With SF} & \text{Training Iterations} & \text{Beat Block (easy)} & \text{Beat Block (hard)}
> \newline \hline
> \times & 3k & 15 & 1
> \newline
> \surd & 3k & 18 & 2
> \newline
> \times & 10k & 35 & 16
> \newline
> \surd & 10k & 42 & 28
> \newline
> \times & 30k & 43 & 21
> \newline
> \surd & 30k & 47 & 34
> \newline
> \times & 100k & 48 & 27
> \newline
> \surd & 100k & 52 & 37
> \newline \hline \end{array}
>
> \begin{array}{cc|cc} \hline
> \text{With SF} & \text{Training Data (episodes)} & \text{Beat Block (easy)} & \text{Beat Block (hard)}
> \newline \hline
> \times & 10 & 9 & 4
> \newline
> \surd & 10 & 22 & 8
> \newline
> \times & 25 & 27 & 9
> \newline
> \surd & 25 & 39 & 22
> \newline
> \times & 50 & 43 & 21
> \newline
> \surd & 50 & 47 & 34
> \newline
> \times & 100 & 47 & 29
> \newline
> \surd & 100 & 52 & 40
> \newline
> \times & 200 & 51 & 36
> \newline
> \surd & 200 & 55 & 48
> \newline \hline \end{array}

---

> > ### Comment · Reviewer_giXr · 2025-11-27
> >
> > Thank you for your response, and I greatly appreciate the additional experiments you have conducted. However, according to the table provided for the additional experiments, the training curve has not converged—thus, it still fails to address my concern: whether the method becomes less effective when the dataset size is large.

---

> > > ### Author Response · Authors · 2025-11-29
> > > **Response to Reviewer giXr**
> > >
> > > **Reply:** We acknowledge that the original data-scaling curves may have caused a misleading impression because the x-axis was displayed in a logarithmic scale. Under the log scale, the curves visually appear much steeper. Therefore, as shown in Figure 13 of the revised Appendix, we provide the data-scaling curves using the linear scale x-axis, which reflects the absolute dataset size without bias. In addition, to make sure that these data-scaling curves fully converged, we conducted further experiments under the larger dataset sizes (the data in the table below are used to plot these data-scaling curves). Experimental results demonstrate that as the dataset size increases, the performance gains gradually become marginal for both the baseline and our SF method. However, we could find that under every data size, our SF consistently outperforms the baseline. In particular, even when the data-scaling curves nearly converge under large data sizes, our SF still maintains a significantly higher success rate. This indicates that our SF remains effective even under large data scales.
> > >
> > > \begin{array}{cc|c} \hline
> > > \text{With SF} & \text{Training Data (episodes)} & \text{Success Rate}
> > > \newline \hline
> > > \times & 10 & 4
> > > \newline
> > > \surd & 10 & 8
> > > \newline
> > > \times & 25 & 9
> > > \newline
> > > \surd & 25 & 22
> > > \newline
> > > \times & 50 & 21
> > > \newline
> > > \surd & 50 & 34
> > > \newline
> > > \times & 100 & 29
> > > \newline
> > > \surd & 100 & 40
> > > \newline
> > > \times & 200 & 36
> > > \newline
> > > \surd & 200 & 48
> > > \newline
> > > \times & 300 & 42
> > > \newline
> > > \surd & 300 & 52
> > > \newline
> > > \times & 400 & 45
> > > \newline
> > > \surd & 400 & 54
> > > \newline
> > > \times & 500 & 46
> > > \newline
> > > \surd & 500 & 54
> > > \newline \hline \end{array}

---

### Official Review · Reviewer_cbE1 · 2025-10-27

**Soundness:** 3
**Presentation:** 3
**Contribution:** 2
**Rating:** 4
**Confidence:** 4

**Summary:**

This paper addresses a critical challenge in Vision-Language-Action (VLA) models: the lack of inherent 3D spatial understanding, which stems from their foundation in Vision-Language Models (VLMs) trained on 2D data. The authors argue that existing solutions, which rely on explicit 3D sensor inputs (e.g., depth, point clouds) or 2D-to-3D estimators, are fraught with practical issues like sensor noise, hardware heterogeneity, data availability, and sub-optimal performance. To overcome this, they propose Spatial Forcing (SF), a novel and simple training strategy. Instead of modifying the model's input, SF implicitly instills spatial awareness by aligning the VLA's intermediate visual representations with geometric features extracted from a powerful, pretrained 3D foundation model (specifically, VGGT). This alignment is achieved via an auxiliary cosine similarity loss during the fine-tuning process. The authors demonstrate through extensive experiments in both simulation and the real world that this implicit supervision method not only achieves state-of-the-art performance, surpassing both 2D and explicit 3D VLA baselines, but also dramatically accelerates training convergence (up to 3.8x) and improves data efficiency.

**Strengths:**

- The paper tackles a fundamental and highly significant problem in embodied AI. The proposed SF method offers an elegant and practical paradigm that sidesteps the many real-world challenges associated with collecting and using explicit 3D data, making it a potentially high-impact contribution for the robotics community.

- The core idea of using representation alignment to implicitly distill 3D knowledge from a foundation model into a VLA is novel and elegant.

- The motivation is exceptionally well-established. The authors' use of a "depth probing" experiment is a simple yet powerful diagnostic tool that compellingly visualizes the problem—the lack of spatial information in standard VLA embeddings—and provides a clear justification for their approach.

- The experimental design is thorough, validating the method across different base models (OpenVLA-OFT, $\pi_0$), diverse simulation environments, and crucial real-world scenarios. The component-wise analysis is also well-executed, providing clear insights into the effects of different target representations and alignment layers.

**Weaknesses:**

- The success of Spatial Forcing is heavily contingent on the availability and quality of a powerful, pretrained 3D foundation model (VGGT in this case). This introduces a strong dependency, and the paper does not discuss the potential limitations if such a model is not available for a specific domain or embodiment, or how the performance of SF scales with the quality of this teacher model.

- While the method is inference-free in terms of overhead, it introduces a non-trivial computational cost during training. It requires running a forward pass through the large VGGT model for every training sample to generate the target representations. This additional overhead in terms of computation and VRAM is not quantified or discussed, which is an important practical consideration that affects the overall "efficiency" claim.

- The paper claims to outperform methods that use explicit 3D inputs (Table 1). While impressive, this comparison may not be entirely fair. SF is effectively distilling knowledge from a very powerful model (VGGT) that has already processed and structured 3D information. In contrast, methods using raw depth maps or point clouds must learn to interpret noisy, unstructured sensor data from scratch. The comparison is thus more akin to "distilled 3D knowledge" vs. "raw 3D data."

- The method uses the "latent representation" from VGGT as the supervision signal. This is somewhat of a black box. It would be beneficial to understand what specific geometric properties (e.g., relative depth, surface normals, object boundaries) are most dominant in these features and are being transferred to the VLA. An ablation using more interpretable outputs from VGGT (like its predicted depth map) as the target could provide valuable insight.

**Questions:**

The paper argues that the aligned representations still preserve their "original representational identity" based on the t-SNE visualization. Could you elaborate on this? How do you ensure that the "forcing" process doesn't cause the visual features to discard important non-spatial information (e.g., texture, color) that might be crucial for other aspects of the task?

---

> ### Author Response · Authors · 2025-11-22
> **Response to Reviewer cbE1 (1/3)**
>
> We deeply appreciate your insightful comments and efforts in reviewing our manuscript. We respond to each of your comments one by one in what follows. In the revised manuscripts, we mark our major revisions as “blue”.
> ### **[W1] The success of Spatial Forcing is heavily contingent on the availability and quality of a powerful, pretrained 3D foundation model (VGGT in this case). This introduces a strong dependency, and the paper does not discuss the potential limitations if such a model is not available for a specific domain or embodiment, or how the performance of SF scales with the quality of this teacher model.**
>
> **Reply:**
> 1. The 3D foundation model (VGGT in this case) is pretrained on large-scale 3D domain data, which is independent of any specific domain or embodiment. Therefore, it can strongly generalize to diverse scenarios and is applicable to different embodiments.
>
> 2. To further address your concern regarding how the performance of SF scales with the quality of the teacher model, we additionally conduct experiments. To the best of our knowledge, VGGT is the first mature 3D foundation model, and we can hardly find another 3D model with a weaker spatial perception ability. Therefore, without loss of generality, we employ a more powerful model than VGGT's multi-view geometric modeling, Depth Anything 3 (DA3)$^{[1]}$, as the teacher and conduct experiments on LIBERO.
>
>     The success rate results, together with our target representation ablations, are listed in the table below. No matter which 3D foundation model is integrated with our SF, the performance improvement is obvious, proving the effectiveness of our SF. In addition, the performance of SF can be improved with a higher quality of the teacher model. Despite this, the marginal improvement from 96.9 to 97.2 indicates that the performance of SF is not significantly affected by the ability of the 3D foundation model.
>
> \begin{array}{c|ccccc} \hline
> \text{Traget Representation} & \text{Spatial} & \text{Object} & \text{Goal} & \text{Long} & \text{Average}
> \newline \hline
> \text{$\times$} & 96.8 & 94.8 & 92.8 & 86.2 & 92.7
> \newline \hline
> \text{VGGT} & 97.2 & 99.2 & 96.8 & 94.2 & 96.9
> \newline
> \text{DA3} & 97.4 & 98.8 & 97.6 & 94.8 & 97.2
> \newline \hline \end{array}
>
> [1] Depth Anything 3: Recovering the Visual Space from Any Views, arxiv 2025.
>
> ### **[W2] While the method is inference-free in terms of overhead, it introduces a non-trivial computational cost during training. It requires running a forward pass through the large VGGT model for every training sample to generate the target representations. This additional overhead in terms of computation and VRAM is not quantified or discussed, which is an important practical consideration that affects the overall "efficiency" claim.**
>
> **Reply:**
> The forward pass of VGGT inevitably introduces additional training overhead. However, for VLAs trained with imitation learning (which includes most mainstream models), the target representations can be gained offline through VGGT's forward pass and stored in advance, thereby not interfering with the normal training process. In addition, we provide the online training overhead on LIBERO as follows:
>
> \begin{array}{c|ccccc} \hline
> \text{Method} & \text{GPU hours} & \text{GPU memory (GB)}
> \newline \hline
> \text{w/o SF} & 29 & 64
> \newline
> \text{w/ SF} & 45 & 73
> \newline \hline \end{array}

---

> ### Author Response · Authors · 2025-11-22
> **Response to Reviewer cbE1 (2/3)**
>
> ### **[W3] The paper claims to outperform methods that use explicit 3D inputs (Table 1). While impressive, this comparison may not be entirely fair. SF is effectively distilling knowledge from a very powerful model (VGGT) that has already processed and structured 3D information. In contrast, methods using raw depth maps or point clouds must learn to interpret noisy, unstructured sensor data from scratch. The comparison is thus more akin to "distilled 3D knowledge" vs. "raw 3D data."**
>
> **Reply:**
> Our method is capable of leveraging the prior knowledge encoded in the powerful 3D foundation model VGGT, whereas other methods that use explicit 3D sensor inputs are unable to take advantage of such foundation models due to their rigid architectural designs and constrained training paradigms. The ability to effectively utilize the spatial perception capabilities of foundation models is itself a key advantage and contribution of our SF strategy. Given an existing foundation model, our approach can seamlessly integrate its strong perceptual representations into the VLA, while others fail to achieve this. **We consider the experiments fair as long as no additional bias (e.g., extra training data or extra privileged information as input) is introduced during training or evaluation that would favor one method over another.**
>
> ### **[W4] The method uses the "latent representation" from VGGT as the supervision signal. This is somewhat of a black box. It would be beneficial to understand what specific geometric properties (e.g., relative depth, surface normals, object boundaries) are most dominant in these features and are being transferred to the VLA. An ablation using more interpretable outputs from VGGT (like its predicted depth map) as the target could provide valuable insight.**
>
> **Reply:**
> We follow your suggestion to design the ablation experiments for a more interpretable analysis below. We begin by introducing the architecture of VGGT, then implement a paradigm for using more interpretable outputs of VGGT as the supervising target, and finally give our results and conclusions.
>
> **VGGT Architecture**. Firstly, let us briefly summarize the VGGT architecture. VGGT first employs global and frame-wise attention layers to obtain unified latent features, which SF uses as target representations. These latent features are then fed into a set of task-specific heads for downstream tasks.
>
> **Implementation.** To understand what specific geometric properties are most dominant in these features, we design a paradigm to directly align VLA with the downstream properties of VGGT. Following the official implementation of VGGT, we mainly choose three geometric properties: depth, point maps, and dynamic tracks. Firstly, we utilize the task-specific heads of VGGT as VLA downstream heads. Then, the visual tokens of VLA are fed into the trainable VLA downstream heads to produce task-specific outputs (i.e., depth, point maps, or dynamic tracks). Finally, we treat the VGGT task-specific outputs as ground-truth and calculate the loss between these two outputs. The overall pipeline is shown in the revised Appendix J.
>
> **Results**. Following the above training pipeline, we train and test our model on the LIBERO-Spatial benchmark. As shown in the table below, when using dynamic point tracks as the supervision signal, the model's performance surpasses that of the other two tasks. This is because it facilitates the modeling of relationships among the end-effector, target objects, and background environment, which is particularly helpful for constraining action-related spatial reasoning. Thus, it is reasonable to assume that **dynamic tracks play a dominant role in transferring geometric properties to the VLA model**. In addition, all variants that directly align with the specific geometric properties lead to bad performance. This is because low-level supervision can disrupt the original visual information of the VLA model, while the VGGT feature serves as a smooth and intermediate supervision to better improve the model's performance.
>
> \begin{array}{c|c|c} \hline
> \text{Supervision Type} & \text{Supervision Signals} & \text{Success Rate}
> \newline \hline
> \text{None}&\times  & 96.8
> \newline \hline
> \text{Specific Geometric Properties}&\text{Depth}  & 84.6
> \newline
> \text{Specific Geometric Properties}&\text{Depth} & 80.0
> \newline
> \text{Specific Geometric Properties}&\text{Point Map} & 87.0
> \newline
> \text{Specific Geometric Properties}&\text{Point Map} & 78.8
> \newline
> \text{Specific Geometric Properties}&\text{Dynamic Tracks}  & 94.2
> \newline
> \text{Specific Geometric Properties}&\text{Dynamic Tracks}  & 85.2
> \newline \hline
> \text{Latent Feature}&\text{VGGT Feature} & 97.2
> \newline \hline \end{array}

---

> ### Author Response · Authors · 2025-11-22
> **Response to Reviewer cbE1 (3/3)**
>
> ### **[Q1] The paper argues that the aligned representations still preserve their "original representational identity" based on the t-SNE visualization. Could you elaborate on this? How do you ensure that the "forcing" process doesn't cause the visual features to discard important non-spatial information (e.g., texture, color) that might be crucial for other aspects of the task?**
>
> **Reply:**
> **From the perspective of VGGT itself:** VGGT itself implements downstream tasks for high-fidelity texture and color reconstruction. For the scene reconstruction task in Figure 3 in the VGGT paper, it successfully reconstructs complex textures, and for novel view synthesis in Section 4.6 in the VGGT paper, it regresses the RGB colors for the target views. **This indicates that VGGT also preserves non-spatial visual features (such as texture and color)** while encoding rich spatial features, and thus the SF alignment process based on VGGT can still retain visual information such as texture.
>
> **From the perspective of our model:** **1. Probing Experiments.** To further validate that the aligned VLA visual tokens still contain non-spatial information (e.g., color), we extend the depth probing experiment in our paper to a color-reconstruction probing experiment. Specifically, we freeze the whole VLA model and train a two-layer MLP and a DPT$^{[2]}$ head to simultaneously predict patch-level color and depth. The probing visualizations are provided in the revised Appendix I, which further confirms the rich color information in the VLA visual tokens after the alignment.
>
> **2. Real-world Experiments.** We additionally evaluate our model on the challenging tasks with **diverse color variation**: grasping the target objects among highly cluttered objects on a table. The target objects include red bell pepper, green bell pepper, red carrot, or green cube. This experimental setting is provided in the revised Appendix H1. Models are trained on 40 demonstrations and evaluated over 40 trials. For each evaluation trial, unseen objects appear on the table and are randomly positioned. The success rate (%) table below demonstrates that **our SF obviously improves the success rates in real-robot scenarios with complex textures and colors**.
>
> \begin{array}{c|c} \hline
> \text{Method} & \text{Success Rate}
> \newline \hline
> \text{w/o SF} & 22.5
> \newline
> \text{w/ SF} & 37.5
> \newline \hline \end{array}
>
> [2] Vision transformers for dense prediction, ICCV 2021.

---

> > ### Comment · Reviewer_cbE1 · 2025-11-25
> > **Official Comment by Reviewer cbE1**
> >
> > Thanks for the authors' response. Most of my concerns have been addressed, I have updated the score to 6.

---

> ### Author Response · Authors · 2025-11-25
> **Response to Reviewer cbE1**
>
> Dear Reviewer cbE1,
>
> We are happy to hear that our response addressed most of your concerns well! Also, we appreciate your support for our work. If you have any further questions or suggestions, please do not hesitate to let us know.
>
> Best regards,
> Authors

---

### Official Review · Reviewer_A7M4 · 2025-11-01

**Soundness:** 3
**Presentation:** 3
**Contribution:** 3
**Rating:** 8
**Confidence:** 4

**Summary:**

The paper presents **Spatial Forcing (SF)**, a training-time regularizer that aligns intermediate visual embeddings of a VLA with geometry-rich representations from a pretrained 3D foundation model (VGGT). This auxiliary alignment loss injects privileged spatial priors into the model during fine-tuning, without requiring explicit 3D inputs or architectural modifications at inference time. Empirically, the authors evaluate SF on **LIBERO** and **RoboTwin 2.0** simulation benchmarks as well as several real-world manipulation tasks. SF consistently improves performance over both 2D and explicit-3D VLAs, achieving up to 3.8× faster convergence and demonstrating robustness to lighting, layout, and height variations in the physical setup.

**Strengths:**

> ### originality
>
> - Addresses the dependency of 3D VLAs on explicit geometric inputs such as depth maps or point clouds.
> - Introduces a simple yet effective co-training mechanism that transfers 3D priors through a pretrained foundation model.
>
> - The privileged-representation intuition is intuitive, and the depth-probing visualization improves interpretability.
>
> ### Quality
>
> - The method is compatible with diverse VLA backbones and requires little engineering effort.
> - The ablation studies on alignment layer and teacher model strengthen the empirical claims

> ### Clarity
>
> - Presents results on LIBERO, RoboTwin 2.0, and real-robot experiments for a comprehensive evaluation.
> - Figures and comparisons are well organized and easy to interpret.
>
> ### Significance
>
> - Bridging 2D vision-based manipulation with 3D perception is an important and rapidly growing research direction.
> - SF provides a practical baseline and may inspire further work on representation alignment and spatial grounding

**Weaknesses:**

- The experiment setting is a bit simple. For simulation the improvement is marginal, while the real world tasks do not demonstrate enough spatial generation in initial pose distribution
- The related work part lacks formal explanation for why representation-level cosine alignment leads to stable 3D awareness.
- The supplemental video includes only few real world experiments, and the tasks are easy with jerky behavior. Please include both simulation and more real world experiments under different noise to prove robustness.
- Currently, the alignment depth layer is chosen empirically, where authors could incorporate an adaptive strategy to improve robustness.

**Questions:**

- In Table 1,the statistics of SpatialVLA, GeoVLA and 3D-CAVLA are same as the origin paper shows, do authors rigorously train different VLAs under same initialization and parameter?
- In Table 2, VGGT without Position Embedding shows high success rates on two Libero benchmark, could you further analyze potential reason?
- In Figure 4, the results are shown as success rates, but there are no explicit percentage labels or variance/error indicators (e.g., standard deviation or confidence intervals). Could the authors clarify whether these bars represent mean performance across seeds or single-run outcomes, and provide variance information to assess statistical significance?
- Did you experiment with multi-layer or progressive alignment instead of a single-layer constraint?
- Can SF improve other vision-driven policies (e.g., diffusion policy or world-model-based RL)?
- Could feature alignment harm language following of base VLAs? Is there a trade-off between spatial and linguistic? Take pi0 experiment for example, can you ablation other task regression?

---

> ### Author Response · Authors · 2025-11-22
> **Response to Reviewer A7M4 (1/3)**
>
> Thank you for your insightful comments and efforts in reviewing our manuscript. We deeply appreciate your valuable suggestions. In the following, we respond to each of your comments point by point. In the revised manuscripts, we mark our major revisions as “blue”.
> ### **[W1] The experiment setting is a bit simple. The improvement is marginal for simulation, while the real-world tasks do not demonstrate enough spatial generation in the initial pose distribution.**
> **Reply:**
> **Simulation**: We politely discuss your concern. As you mentioned, LIBERO is a relatively simple benchmark with limited task diversity, where most mainstream models already achieve near-saturated success rates. Although our method brings only a modest improvement to the baseline under sufficient training steps and abundant training data (success rate: 97.1% -> 98.5%), it delivers crucial gains in training efficiency (3.8x) and data efficiency (5.9x) when training steps and data are limited.
>
> As we have already reported in the original manuscript, when evaluated on the more challenging RoboTwin benchmark, we show that the improvement is quite significant, with success rates rising from 22.1% to 31.3% in the hard setting.
>
> **Real-world Experiments**: Regarding the real-world experiments under random initial pose distribution, we additionally conduct evaluations on our three single-arm tasks with variation. We test 40 trials for each single-arm task, and the resulting success rates (%) are summarized in the table below. The results demonstrate that our approach still exhibits improved spatial capabilities in out-of-distribution (OOD) initial pose distribution scenarios.
>
> \begin{array}{c|ccc} \hline  \text{Method} & \text{Stack Glass Cups} & \text{Grasp Right-side Vegetable} & \text{Place Green Block}\newline \hline \text{w/o SF} & 5 & 5 & 22.5 \newline \text{w/ SF} & 27.5 & 17.5 & 40 \newline \hline \end{array}
>
> ### **[W2] The related work part lacks formal explanation for why representation-level cosine alignment leads to stable 3D awareness.**
> **Reply:**
> Following REPA$^{[1]}$ and Geometry Forcing$^{[2]}$, the cosine similarity loss aligns hidden states at the pixel-patch level independently. Owing to the global attention across patches within the VGGT backbone, this alignment loss yields a stable 3D feature representation for the whole scene.
>
> [1] Representation Alignment for Generation: Training Diffusion Transformers Is Easier Than You Think, ICLR 2025.
>
> [2] Geometry Forcing: Marrying Video Diffusion and 3D Representation for Consistent World Modeling, arxiv 2025.
>
> ### **[W3] The supplemental video includes only few real world experiments, and the tasks are easy with jerky behavior. Please include both simulation and more real world experiments under different noise to prove robustness.**
> **Reply:**
> Following your instruction, in the revised supplementary material, we provide more **simulation videos under domain-randomization** and **real-world videos under noise and variations**. Specifically, for the simulated environment, we include RoboTwin simulation videos under domain-randomized clutter, lighting, textures, and table-height variations, demonstrating the robustness of our approach. For the real-world experiments, we train a single model to handle all variations in one task, indicating our robustness in the real world.
>
> *Jerky behavior.* The jerky behavior is primarily caused by the robot control frequency and the mechanical properties of the arm motors. To address this issue, we interpolate the predicted actions to the higher control frequency of 200 Hz, as required by the official setup, which substantially mitigates the jerky effect.

---

> ### Author Response · Authors · 2025-11-22
> **Response to Reviewer A7M4 (2/3)**
>
> ### **[W4] Currently, the alignment depth layer is chosen empirically, where authors could incorporate an adaptive strategy to improve robustness.**
> **Reply:**
> Thank you for your valuable suggestion. Following your suggestion, we **developed an adaptive strategy** to automatically choose the alignment depth layer of VLAs, and we **found this strategy is effective**.
>
> Specifically, following the core idea of MOE soft gating mechanism$^{[3, 4]}$, firstly we employ an trainable weight matrix $M _ g \in \mathbb{R}^{L\times D}$, where $L$ is the total amount of VLA layers (e.g., 32 for LLaVA.), $D$ is the feature dimension.
>
> Then, pool the VLAs' visual tokens along the token axis: $P=\frac{1}{N}\sum _ {\mathcal{t}=1}^{N} x^\mathcal{V} _ \mathcal{t}$, where $\lbrace x^\mathcal{V} _ \mathcal{t} \rbrace _ {t=1}^{N}$ is N visual token of VLAs. If the visual tokens ${}^lx^\mathcal{V}$ are gained at the $l _ {th}$ layer of VLAs, we denote the corresponding pooling feature as $P(l)$, where $l=\lbrace 1,2, ..., L \rbrace $.
>
> Next, calculate the gating score scalar for each layer: $s _ l = M _ g(l)P(l)^\texttt{T}$.
>
> Finally, we use the mixtual visual feature to calculate the align loss: $\mathcal{F} _ {align}[\sum _ {l=1}^L\frac{\texttt{exp}(s _ l)}{\sum _ {j=1}^L\texttt{exp}(s _ j)}{}^lx^\mathcal{V}, f^{3D}]$.
>
> We report the updated success rate on LIBERO. As shown in the table below, the adaptive layer selection strategy shows clear improvements. And we provide the gating score distribution histograms across layers in revised Appendix G.
>
> \begin{array}{c|ccccc} \hline  \text{Aligned Layer$^{th}$} & \text{Spatial} & \text{Object} & \text{Goal} & \text{Long} & \text{Average}\newline \hline \text{1} & 96.8 & \textbf{99.4} & \textbf{99.0} & 83.0 & 94.6 \newline \text{8} & 96.2 & 98.4 & 95.6 & 92.4 & 95.7 \newline \text{16} & 97.4 & 98.8 & 95.8 & 83.2 & 93.8 \newline \text{24} & 97.2 & 99.2 & 96.8 & \underline{94.2} & \underline{96.9} \newline \text{32} & \textbf{98.8} & \textbf{99.4} & 96.2 & 84.8 & 94.8 \newline \text{adaptive} & \underline{98.6} & \textbf{99.4} & \underline{98.8} & \textbf{95.4} & \textbf{98.1} \newline \hline \end{array}
>
> [3] Outrageously large neural networks: The sparsely-gated mixture-of-experts layer, ICLR 2017.
>
> [4] Selective Kernel Networks, CVPR 2019.
>
> ### **[Q1] In Table 1,the statistics of SpatialVLA, GeoVLA and 3D-CAVLA are same as the origin paper shows, do authors rigorously train different VLAs under same initialization and parameter?**
> **Reply:**
> Because the SpatialVLA and GeoVLA do not release the training codes on LIBERO, we mainly train 3D-CAVLA under the same initialization and parameters. We find that under these settings, the results of 3D-CAVLA remain almost consistent with the original reported results.
>
> ### **[Q2] In Table 2, VGGT without Position Embedding shows high success rates on two Libero benchmark, could you further analyze potential reason?**
> **Reply:**
> As LIBERO represents a relatively simple benchmark, the success rates for the first three subtasks (Spatial, Object, and Goal) in the positional embedding ablation study (Table 2 in the main manuscript) are already near saturation. Consequently, marginal variances of less than 1% are insufficient to discern genuine performance improvements. In contrast, the fourth subtask (Long) provides a more rigorous assessment of the advantage of positional encoding, thereby making the average success rate a reliable indicator of performance enhancement (improving from 94.7% to 96.9%).
>
> ### **[Q3] In Figure 4, the results are shown as success rates, but there are no explicit percentage labels or variance/error indicators (e.g., standard deviation or confidence intervals). Could the authors clarify whether these bars represent mean performance across seeds or single-run outcomes, and provide variance information to assess statistical significance?**
> **Reply:**
> Our reported success rate bars in Figure 4 represent the mean performance across **three random seeds**. For each seed, we follow the official evaluation protocol of RoboTwin2.0 to test models with 100 rollouts. Below are the success rate (%) of the SF model, together with **standard deviations**, on the RoboTwin hard tasks.
>
> \begin{array}{cccccccc} \hline
> \text{Move Playingcard} & \text{Turn Switch} & \text{Click Bell} & \text{Open Microwave} & \text{Beat Block} & \text{Lift Pot} & \text{Place Shoes}
> \newline \hline
> 32.0_{\pm 3.6} & 23.3_{\pm 5.5} & 8.0_{\pm 3.6} & 76.7_{\pm 3.1} & 34.0_{\pm 1.6} & 42.4_{\pm 2.0} & 3.3_{\pm 1.5}
> \newline \hline \end{array}

---

> ### Author Response · Authors · 2025-11-22
> **Response to Reviewer A7M4 (3/3)**
>
> ### **[Q4] Did you experiment with multi-layer or progressive alignment instead of a single-layer constraint?**
> **Reply:**
> If we understand correctly, this question is relevant to the [W4]. Please refer to our reply to [W4].
>
> ### **[Q5] Can SF improve other vision-driven policies (e.g., diffusion policy or world-model-based RL)?**
> **Reply:**
> To validate whether our SF strategy is effective for other vision-driven policies, we employ the representative method, ACT$^{[5]}$, as the baseline and conduct experiments on the RoboTwin simulator.
>
> **Implementation.** Specifically, the ACT utilizes a transformer encoder to convert images from multiple viewpoints, joint positions, and a style variable $z$ into a series of enhanced tokens. We take out the corresponding enhanced image tokens and align them with external VGGT representations.
>
> **Results.** As shown in the result table below, the ACT enhanced with SF achieves higher success rates across different tasks and settings. This demonstrates that the SF strategy is not limited to VLA paradigms but is also effective on other driven policies. Particularly, for the *move playcard (hard)* task, the playcard box is so thin that it only protrudes slightly from the table surface, making the ACT baseline fail to grasp. By enhancing the spatial perception precision, SF achieves success cases.
>
> \begin{array}{c|cccc} \hline
> \text{Method} & \text{Beat Block (easy)} & \text{Beat Block (hard)} & \text{Move Playcard (easy)} & \text{Move Playcard (hard)}
> \newline \hline
> \text{ACT (offical)} & 56 & 3 & 36 & 0
> \newline
> \text{ACT (reproduced)} & 52 & 3 & 41 & 0
> \newline
> \text{SF (ours)} & 61 & 7 & 52 & 8
> \newline \hline \end{array}
>
> [5] Learning Fine-Grained Bimanual Manipulation with Low-Cost Hardware, RSS 2023.
>
> ### **[Q6] Could feature alignment harm language following of base VLAs? Is there a trade-off between spatial and linguistic? Take pi0 experiment for example, can you ablation other task regression?**
> **Reply:**
> The factors determining whether a VLA successfully completes a task involve both following human instructions and precise action execution. Let us take the "beat block" task on the RoboTwin benchmark as an example. The agent succeeds in instruction following if it successfully chooses the correct direction to move close to the hammer and the red block, whereas the agent succeeds in execution if it successfully picks up the hammer and hits the block.
>
> In practice, we manually check all the task rollout videos to statistics the instruction-following and execution success rate, and list them separately in the table below. It can be observed that the baseline and SF have almost the same instruction-following success rate. Therefore, we conclude that SF does not harm the language following of base VLAs. Part of the comparison videos is attached in the revised supplementary material.
>
> \begin{array}{c|cc} \hline
> \text{Method} & \text{Instruction-follwing} & \text{Excution} & \text{Overall}
> \newline \hline
> \text{Baseline} & 59.0 & 35.6 & 21.0
> \newline
> \text{SF (ours)} & 60.3 & 56.0 & 34.0
> \newline \hline \end{array}

---

### Author Response · Authors · 2025-12-01
**Summary of Rebuttal**

Dear Area Chair and Reviewers,

We sincerely appreciate your time and valuable feedback. Given the recent reassignment of the AC, we offer a brief summary of our submission during rebuttal to help the AC's final decision.

- Reviewer cbE1 increased the score from 4 to **6** on Nov 24 (AoE), before the OpenReview bug became publicly known. This is also supported by this reviewer's comment.
- Reviewer giXr, who assigned a score of **6**, greatly appreciated the experiments we provided in the first round of response and required us to further clarify the scaling issue. As a response, we conducted the relevant experiments, which we believe have substantially resolved this reviewer's concern. We did not receive a further reply from this reviewer due to the suspension of reviewers' comments.
- Reviewer A7M4 assigned a score of **8** and Reviewer UKHb assigned a score of **6**. However, they did not respond to our update before the reviewers' comments were suspended. We believe that we have substantially resolved the concerns raised by these two reviewers.

Sincerely,

The Authors

---

### Meta-Review · Area_Chair_xveS · 2026-01-11

**Summary:**

This paper introduces Spatial Forcing (SF), a training-time alignment regularizer for vision-language-action (VLA) models that aims to improve 3D spatial awareness without requiring explicit 3D inputs. Reviewers agree the paper targets an important limitation of VLAs and propose an elegant, easy-to-integrate method with strong empirical results and ablations. While concerns about teacher dependence, training overhead, and evaluation rigor are valid, most of them have been addressed during rebuttal. After carefully reading the paper, review and author responses, the AC agrees with the majority of the reviewers on accepting the paper.

**Reviewer Concerns:**

see Summary

**Reviewer Scores:**

see Summary

---

### Decision · Program_Chairs · 2026-01-26

Accept (Poster)